# Self-Supervised Graph Neural Networks for Improved Electroencephalographic Seizure Analysis

**Siyi Tang, Jared Dunnmon, Khaled Saab, Xuan Zhang**[†]**, Qianying Huang**[†]**, Florian Dubost, Daniel Rubin**[‡]**, Christopher Lee-Messer**[‡]

Stanford University, CA, USA

`{siyitang,jdunnmon,ksaab,kayleez,qyhuang,fdubost,rubin,cleemess}`
`@stanford.edu`

[†] Equal contributions. [‡] Equal contributions.

## Abstract

Automated seizure detection and classification from electroencephalography (EEG) can greatly improve seizure diagnosis and treatment. However, several modeling challenges remain unaddressed in prior automated seizure detection and classification studies: (1) representing non-Euclidean data structure in EEGs, (2) accurately classifying rare seizure types, and (3) lacking a quantitative interpretability approach to measure model ability to localize seizures. In this study, we address these challenges by (1) representing the *spatiotemporal dependencies* in EEGs using a graph neural network (GNN) and proposing two EEG graph structures that capture the *electrode geometry* or *dynamic brain connectivity*, (2) proposing a *self-supervised pre-training* method that predicts preprocessed signals for the next time period to further improve model performance, particularly on rare seizure types, and (3) proposing a *quantitative model interpretability* approach to assess a model's ability to localize seizures within EEGs. When evaluating our approach on seizure detection and classification on a large public dataset (5,499 EEGs), we find that our GNN with self-supervised pre-training achieves 0.875 Area Under the Receiver Operating Characteristic Curve on seizure detection and 0.749 weighted F1-score on seizure classification, outperforming previous methods for both seizure detection and classification. Moreover, our self-supervised pre-training strategy significantly improves classification of rare seizure types (e.g. 47 points increase in combined tonic seizure accuracy over baselines). Furthermore, quantitative interpretability analysis shows that our GNN with self-supervised pre-training precisely localizes 25.4% focal seizures, a 21.9 point improvement over existing CNNs. Finally, by superimposing the identified seizure locations on both raw EEG signals and EEG graphs, our approach could provide clinicians with an intuitive visualization of localized seizure regions.

## 1 Introduction

Seizures are among the most common neurological emergencies in the world (Strein et al., 2019). Seizures can be chronic as in the case of epilepsy, a neurological disease affecting 50 million people worldwide (WHO, 2019). Clinically, definitive detection of a seizure is only the first step in seizure diagnosis. An important subsequent step is to classify seizures into finer-grained types—such as focal versus generalized seizures—for identifying epilepsy syndromes, targeted therapies, and eligibility for epilepsy surgery (Fisher et al., 2017).

Scalp electroencephalography (or "EEG") plays a critical role in seizure detection and classification. Clinically, EEG-based seizure detection and classification are performed by a trained EEG reader who visually examines a patient's EEG signals over time periods ranging from hours to days. However, this manual analysis is extremely resource- and time-intensive, and thus automated algorithms could greatly accelerate seizure diagnosis and improve outcomes for severely ill patients.

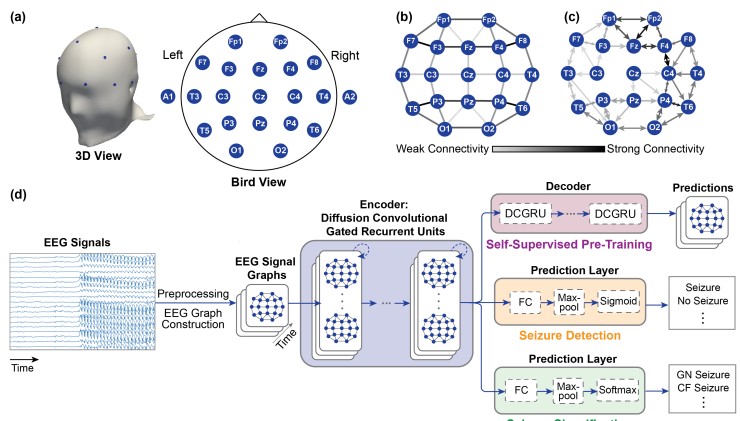

Figure 1: **(a)** EEG electrode placement in the standard 10-20 system. **(b)** Distance-based EEG graph. **(c)** An example correlation-based EEG graph. **(d)** Overview of our methods. The inputs to the models are the EEG graphs, where each node feature corresponds to the preprocessed EEG signals in the respective channel. Self-edges are not shown for better visualization.

Although a large number of studies have attempted automated seizure detection (Rasheed et al., 2020; Siddiqui et al., 2020; Shoeibi et al., 2021; O'Shea et al., 2020; Saab et al., 2020) or seizure classification (Raghu et al., 2020; Asif et al., 2020; Iesmantas & Alzbutas, 2020; Ahmedt-Aristizabal et al., 2020; Roy et al., 2019), several challenges remain largely unaddressed. First, most recent studies use convolutional neural networks (CNNs) that assume Euclidean structures in EEG signals or spectrograms (Rasheed et al., 2020; Shoeibi et al., 2021; Raghu et al., 2020; Asif et al., 2020; Iesmantas & Alzbutas, 2020; Ahmedt-Aristizabal et al., 2020; Roy et al., 2019; O'Shea et al., 2020; Saab et al., 2020). However, assumption of Euclidean structure ignores the natural geometry of EEG electrodes and the connectivity in brain networks. EEGs are measured by electrodes placed on a manifold (i.e. patient's scalp) (Figure 1a), and thus have inherent non-Euclidean structures. Graphs are a data structure that can represent complex, non-Euclidean data (Chami et al., 2020; Bronstein et al., 2017), and graph theory has been extensively used in modeling brain networks (Bullmore & Sporns, 2009). We therefore hypothesize that graph-based modeling approaches can better represent the inherent non-Euclidean structures in EEGs in a manner that improves both the performance and the clinical utility of seizure detection and classification models. Although traditional graph theory has been used (Supriya et al., 2021), only a few deep learning studies have modeled EEGs as graphs for seizure detection. However, these graph-based studies were limited to nonpublic (Covert et al., 2019) or small datasets (Craley et al., 2019; Li et al., 2021), did not leverage modern self-supervised approaches or examine seizure classification (Cisotto et al., 2020; Zhao et al., 2021; Li et al., 2021).

Second, certain seizure types (e.g. clonic seizures) are rare by nature. Training machine learning models that perform well on these rarer seizure classes using traditional supervised learning approaches is challenging, which could explain the performance difference between major and minority seizure types in prior studies (Raghu et al., 2020; Iesmantas & Alzbutas, 2020; Ahmedt-Aristizabal et al., 2020). Several studies have investigated an alternative, self-supervised training strategy (Banville et al., 2020; Mohsenvand et al., 2020; Kostas et al., 2021; Martini et al., 2021; Xu et al., 2020), but they did not model EEGs as graphs or address automated seizure classification. Prior works have shown that self-supervised pre-training significantly improves model performance on data with imbalanced labels in the field of computer vision (Yang & Xu, 2020; Liu et al., 2021). Hence, we hypothesize that self-supervised pre-training can help improve our graph model performance on rare seizure types. Moreover, a large portion of EEG signals generally do not have seizures. Self-supervised pre-training strategy would allow the model to leverage the abundant non-seizure EEGs that are readily available in the dataset.

Lastly, for seizure detection and classification models, the ability to not only provide a single prediction across all EEG channels, but also to provide the interpretability and the ability to localize seizures would be clinically useful for informing treatment strategy. While prior studies (Saab et al., 2020; Covert et al., 2019) have shown qualitative visualization for model interpretability, none have quantitatively assessed the model's ability to localize seizures.

In this work, we aim to address these limitations in prior automated seizure detection and classification studies. First, we propose a graph-based modeling approach for EEG-based seizure detection and classification. Specifically, we propose two EEG graph structures that capture EEG sensor

geometry (Figure 1b) or dynamic brain connectivity (Figure 1c), and we extend Diffusion Convolutional Recurrent Neural Network (DCRNN) (Li et al., 2018), an RNN with graph diffusion convolutions, to model the spatiotemporal dependencies in EEGs (Figure 1d). Second, we improve DCRNN performance using a self-supervised pre-training strategy of predicting the preprocessed EEG signals for the next time period without requiring additional data or labels. Finally, we propose quantitative metrics to assess our model's ability to localize seizures. In summary:

- We propose two EEG graph structures that capture (1) the *natural geometry* of EEG sensors or (2) *dynamic connectivity* in the brain, and show that building a recurrent graph neural network (GNN) based on these representations yields models for seizure detection and classification that outperform previous approaches on a large public dataset (5,499 EEGs).

- We propose a *self-supervised pre-training* strategy to further improve our recurrent GNN model performance, particularly on rare seizure types. To our knowledge, our study is the first to date that combines graph-based modeling and self-supervised pre-training for EEGs. By leveraging graph structure and self-supervision, our method achieves 0.875 Area Under the Receiver Operating Characteristic Curve (AUROC) on seizure detection and 0.749 weighted F1-score on seizure classification, outperforming previous approaches on both seizure detection and classification on this large public dataset. Moreover, our self-supervised pre-training method substantially improves classification of rare seizure types (e.g. 47 points increase in combined tonic seizure accuracy over baselines).

- We propose a *quantitative model interpretability* analysis that can be used to assess a model's ability to localize seizures, which is critical to determining the course of treatment for seizures. We show that by leveraging graph structure and self-supervision, our method precisely localizes 25.4% focal seizures, providing 21.9 points improvement over a prior state-of-the-art CNN. Finally, by displaying the identified seizure regions on raw EEG signals and EEG graphs, our approach could provide valuable insights that support more effective seizure diagnosis in real-world clinical settings.

## 2 METHODS

### 2.1 SEIZURE DETECTION AND CLASSIFICATION PROBLEM FORMULATION

The goal of seizure detection is to predict whether a seizure exists within an EEG clip, and the goal of seizure classification is to predict the seizure type given a seizure EEG clip. Following a prior study (Saab et al., 2020), we examine our model's capability for fast and slow detection and classification over 12-s and 60-s EEG clips, respectively.

### 2.2 GRAPH-BASED MODELING FOR EEGS

#### 2.2.1 REPRESENTING EEGS AS GRAPHS

We represent an EEG clip as a graph $\mathcal{G} = \{\mathcal{V}, \mathcal{E}, \boldsymbol{W}\}$, where $\mathcal{V}$ denotes the set of nodes (i.e. EEG electrodes/channels), $\mathcal{E}$ denotes the set of edges, and $\boldsymbol{W}$ is the adjacency matrix. We propose the following two methods of constructing the EEG graph.

**Distance graph.** To represent the *natural geometry* of EEG electrodes, we compute edge weight $\boldsymbol{W}_{ij}$ by applying a thresholded Gaussian kernel (Shuman et al., 2013) to the pairwise Euclidean distance between $v_i$ and $v_j$, i.e., $\boldsymbol{W}_{ij} = \exp\left(-\dfrac{\text{dist}(v_i, v_j)^2}{\sigma^2}\right)$ if $\text{dist}(v_i, v_j) \leq \kappa$, else 0. Here, $\text{dist}(v_i, v_j)$ is the Euclidean distance between electrodes $v_i$ and $v_j$ according to the standard 10-20 EEG electrode placement (Jasper, 1958), $\sigma$ is the standard deviation of the distances, and $\kappa$ is the threshold for sparsity. This results in a universal undirected, weighted graph for all EEG clips. Based on preliminary experiments and EEG domain knowledge, we chose $\kappa = 0.9$ because it results in a reasonable graph that also resembles the EEG montage (longitudinal bipolar and transverse bipolar) widely used clinically (Acharya et al., 2016). Figure 1b shows the distance graph with $\kappa = 0.9$. In Appendix K, we explore different values of $\kappa$ as well as constructing the distance graph using a Gaussian kernel with a pre-specified bandwidth.

**Correlation graph.** To capture *dynamic brain connectivity*, we define the edge weight $\boldsymbol{W}_{ij}$ as the absolute value of the normalized cross-correlation between the preprocessed signals in $v_i$ and $v_j$. To introduce sparsity to the graph, only the edges whose weights are among the top-$\tau$ neighbors of each node are kept (plus self-edges), i.e., $\boldsymbol{W}_{ij} = |\boldsymbol{X}_{:,i,:} * \boldsymbol{X}_{:,j,:}|$ if $v_j \in \mathcal{N}(v_i)$, else 0. Here, $\boldsymbol{X}_{:,i,:}$ and $\boldsymbol{X}_{:,j,:}$ are preprocessed signals in $v_i$ and $v_j$, $*$ represents the normalized cross-correlation, and $\mathcal{N}(v_i)$ represents the top-$\tau$ neighbors of $v_i$. This method results in a unique directed, weighted graph for each input EEG clip. Figure 1c shows an example correlation graph with $\tau = 3$.

### 2.2.2 GRAPH NEURAL NETWORK

We adapt DCRNN (Li et al., 2018), a recurrent neural network with graph diffusion convolutions, to model the *spatiotemporal dependencies* in EEG signals. DCRNN was initially developed for traffic forecasting, where the dynamics of traffic flow are modeled as a diffusion process. Similarly, we can also model the *spatial dependency* in EEG signals as a diffusion process, because an electrode can be influenced more by electrodes in its anatomical proximity (measured by distance) (Acharya et al., 2016) or functional proximity (measured by correlation) (Sakkalis, 2011). Specifically, the diffusion process is characterized by a bidirectional random walk on a directed graph $\mathcal{G}$, which results in the following diffusion convolution (Li et al., 2018):

$$\boldsymbol{X}_{:,m} \star_{\mathcal{G}} f_\theta = \sum_{k=0}^{K-1} \left( \theta_{k,1} (\boldsymbol{D}_O^{-1} \boldsymbol{W})^k + \theta_{k,2} (\boldsymbol{D}_I^{-1} \boldsymbol{W}^{\mathsf{T}})^k \right) \boldsymbol{X}_{:,m} \text{ for } m \in \{1, ..., M\} \quad (1)$$

where $\boldsymbol{X} \in \mathbb{R}^{N \times M}$ is the preprocessed EEG clip at time step $t \in \{1, ..., T\}$ with $N$ nodes and $M$ features, $f_\theta$ is the convolution filter with parameters $\theta \in \mathbb{R}^{K \times 2}$, $\boldsymbol{D}_O$ and $\boldsymbol{D}_I$ are the out-degree and in-degree diagonal matrices of the graph, respectively, $\boldsymbol{D}_O^{-1} \boldsymbol{W}$ and $\boldsymbol{D}_I^{-1} \boldsymbol{W}^{\mathsf{T}}$ are the state transition matrices of the outward and inward diffusion processes, respectively, and $K$ is the number of maximum diffusion steps.

For undirected graphs, the diffusion convolution is similar to ChebNet spectral graph convolution (Defferrard et al., 2016) up to a constant scaling factor, and thus can be computed using stable Chebyshev polynomial bases as follows (Li et al., 2018):

$$\boldsymbol{X}_{:,m} \star_{\mathcal{G}} f_\theta = \boldsymbol{\Phi} \left( \sum_{k=0}^{K-1} \theta_k \boldsymbol{\Lambda}^k \right) \boldsymbol{\Phi}^{\mathsf{T}} \boldsymbol{X}_{:,m} = \sum_{k=0}^{K-1} \theta_k \boldsymbol{L}^k \boldsymbol{X}_{:,m} = \sum_{k=0}^{K-1} \tilde{\theta}_k T_k(\tilde{\boldsymbol{L}}) \boldsymbol{X}_{:,m} \text{ for } m \in \{1, ..., M\}$$
(2)

where $T_0(x) = 1, T_1(x) = x$, and $T_k(x) = 2x T_{k-1}(x) - T_{k-2}(x)$ for $k \geq 2$ are the bases of the Chebyshev polynomial, $\boldsymbol{L} = \boldsymbol{D}^{-\frac{1}{2}} (\boldsymbol{D} - \boldsymbol{W}) \boldsymbol{D}^{-\frac{1}{2}} = \boldsymbol{\Phi} \boldsymbol{\Lambda} \boldsymbol{\Phi}^{\mathsf{T}}$ is the normalized graph Laplacian, and $\tilde{\boldsymbol{L}} = \frac{2}{\lambda_{max}} \boldsymbol{L} - \boldsymbol{I}$ is the scaled graph Laplacian mapping eigenvalues from $[0, \lambda_{max}]$ to $[-1, 1]$. We use Equation 1 for directed correlation graphs, and Equation 2 for undirected distance graph.

Next, to model the *temporal dependency* in EEGs, we employ Gated Recurrent Units (GRUs) (Cho et al., 2014), a variant of RNN with a gating mechanism. Specifically, the matrix multiplications in GRUs are replaced with diffusion convolutions (or ChebNet spectral graph convolutions for undirected distance-based graph) (Li et al., 2018), allowing *spatiotemporal modeling* of EEG signals (referred to as "DCGRU"):

$$\boldsymbol{r}^{(t)} = \sigma \left( \boldsymbol{\Theta}_r \star_{\mathcal{G}} [\boldsymbol{X}^{(t)}, \boldsymbol{H}^{(t-1)}] + \mathbf{b}_r \right) \qquad \boldsymbol{u}^{(t)} = \sigma \left( \boldsymbol{\Theta}_u \star_{\mathcal{G}} [\boldsymbol{X}^{(t)}, \boldsymbol{H}^{(t-1)}] + \mathbf{b}_u \right) \quad (3)$$

$$\boldsymbol{C}^{(t)} = \tanh \left( \boldsymbol{\Theta}_C \star_{\mathcal{G}} [\boldsymbol{X}^{(t)}, (\boldsymbol{r}^{(t)} \odot \boldsymbol{H}^{(t-1)})] + \mathbf{b}_C \right) \quad \boldsymbol{H}^{(t)} = \boldsymbol{u}^{(t)} \odot \boldsymbol{H}^{(t-1)} + (1 - \boldsymbol{u}^{(t)}) \odot \boldsymbol{C}^{(t)} \quad (4)$$

Here, $\boldsymbol{X}^{(t)}$, $\boldsymbol{H}^{(t)}$ denote the input and output of DCGRU at time step $t$ respectively, $\sigma$ denotes Sigmoid function, $\odot$ represents the Hadamard product, $\boldsymbol{r}^{(t)}, \boldsymbol{u}^{(t)}, \boldsymbol{C}^{(t)}$ denote reset gate, update gate and candidate at time step $t$ respectively, $\star_{\mathcal{G}}$ denotes the diffusion convolution (or ChebNet spectral graph convolution), $\boldsymbol{\Theta}_r, \mathbf{b}_r, \boldsymbol{\Theta}_u, \mathbf{b}_u, \boldsymbol{\Theta}_C$ and $\mathbf{b}_C$ are the weights and biases for the corresponding convolutional filters. Finally, for seizure detection and classification, the models consist of several stacked DCGRUs followed by a fully-connected layer.

### 2.3 SELF-SUPERVISED PRE-TRAINING

To further improve DCRNN performance, we propose a self-supervised pre-training strategy for EEGs. Specifically, we pre-train the model for predicting the next $T'$ second preprocessed EEG

clips given a preprocessed 12-s (60-s) EEG clip. We hypothesize that by learning to predict the EEG signals for the next time period, the model would learn task-agnostic representations and improve downstream seizure detection and classification tasks. The model for self-supervised pre-training has a sequence-to-sequence architecture with an encoder and a decoder (Sutskever et al., 2014), each of which has several stacked DCGRUs (Figure 1d). We use mean absolute error between the true preprocessed EEG clips and the predicted clips as the loss function. Preliminary experiments suggest that predicting future $T' = 12$ second preprocessed EEG clips results in low regression loss on the validation set, and thus we use $T' = 12$ in all self-supervised pre-training experiments.

## 2.4 MODEL INTERPRETABILITY AND ABILITY TO LOCALIZE SEIZURES

We perform model interpretability analyses using an occlusion-based approach (Zeiler & Fergus, 2013). First, for seizure detection, we zero-fill one second EEG signals in one channel at a time and compute the relative change in the model's output logit with respect to the original non-occluded output (original − occluded). Furthermore, we scale the values in each clip to $[0, 1]$ using the minimum and maximum values in that clip. This results in an occlusion map $M \in \mathbb{R}^{N \times T}$, where $N$ is the number of EEG channels, $T$ is the clip length, and $M_{ij}$ indicates the relative change in the model output when the $j$-th second EEG clip in the $i$-th channel is occluded. A larger $M_{ij}$ indicates that the occluded region is more important for predicting seizure, and vice versa. We visualize $M$ by superimposing it over raw EEG signals and graph structures.

In addition to visually analyzing the aforementioned occlusion maps, we propose quantitative metrics to evaluate the model's capability of localizing seizures based on the occlusion maps. Specifically, we define a coverage metric that quantifies how many true seizure regions are detected, as well as a localization metric that quantifies how many detected seizure regions are true seizure regions. Note that coverage and localization scores are analogous to recall and precision, respectively, in binary classification problems. Mathematically, let $M^{\text{annot}} \in \mathbb{R}^{N \times T}$ be detailed annotations of seizure duration and location, where $M_{ij}^{\text{annot}} = 1$ if there is a seizure at the $j$-th second in the $i$-th channel, otherwise $M_{ij}^{\text{annot}} = 0$. Let $M \in \mathbb{R}^{N \times T}$ be the occlusion map described above. Then

$$\text{coverage} = \frac{\sum_{i,j} \mathbf{1}_{M_{ij} > 0.5} M_{ij}^{\text{annot}}}{\sum_{i,j} M_{ij}^{\text{annot}}} \text{ and localization} = \frac{\sum_{i,j} \mathbf{1}_{M_{ij} > 0.5} M_{ij}^{\text{annot}}}{\sum_{i,j} \mathbf{1}_{M_{ij} > 0.5}}.$$

Finally, we also perform occlusion-based interpretability analysis for seizure classification. Since the goal of seizure classification is to predict the seizure type given a seizure EEG clip, we hypothesize that the difference in signals between EEG channels are more important for predicting seizure types. Therefore, we completely drop one EEG channel at a time, and compute the relative change in the model output with respect to the original output. This results in an occlusion map $M' \in \mathbb{R}^N$, where $M'_i$ indicates the relative change in the model output when the $i$-th channel is dropped.

## 3 EXPERIMENTS

### 3.1 EXPERIMENTAL SETUP

**Dataset.** We use the public Temple University Hospital EEG Seizure Corpus (TUSZ) v1.5.2 (Shah et al., 2018; Obeid & Picone, 2016), the largest public EEG seizure database to date with 5,612 EEGs, 3,050 annotated seizures from clinical recordings, and eight seizure types[1]. We include 19 EEG channels in the standard 10-20 system (Figure 1b–1c). To evaluate model generalizability to unseen patients, we exclude five patients from the official TUSZ test set who exist in both the official TUSZ train and test sets. Moreover, we use the detailed annotations of seizure duration and location available in TUSZ for our interpretability analyses. Table 1 summarizes the TUSZ data.

**Data preprocessing.** Because seizures are associated with brain electrical activity in certain frequency bands (Tzallas et al., 2009), we perform data preprocessing to transform raw EEG signals to the frequency domain. Similar to prior studies (Asif et al., 2020; Ahmedt-Aristizabal et al., 2020; Covert et al., 2019), we obtain the log-amplitudes of the fast Fourier transform of raw EEG signals. For seizure detection and self-supervised pre-training, we use both seizure and non-seizure

---

[1] The eight seizure types are: focal, generalized non-specific, simple partial, complex partial, absence, tonic, tonic-clonic, and myoclonic seizures.

EEGs, and obtain the 12-s (60-s) EEG clips using non-overlapping 12-s (60-s) sliding windows. For seizure classification, we use only seizure EEGs, and obtain one 12-s (60-s) EEG clip from each seizure event, such that each EEG clip has exactly one seizure type. Appendix A presents details of data preprocessing, and Appendix B compares results of frequency-domain vs time-domain inputs.

**Refined seizure classification scheme.** Because simple partial (SP) seizures and complex partial (CP) seizures are focal seizures characterized by a clinical behavior (consciousness during a seizure) and are not distinguishable by EEG signals alone (Fisher et al., 2017), we combine focal non-specific (FN), SP, and CP seizures to form a combined focal (CF) seizure class. We provide extensive experiments in Appendix C showing that these focal seizure types cannot be distinguished by a variety of models. We also exclude myoclonic seizures because only three myoclonic seizures are available in TUSZ. In addition, since tonic and tonic-clonic seizures are rare in the dataset (only 18 tonic seizures and 30 tonic-clonic seizures in TUSZ train set), and tonic-clonic seizures always start with a tonic phase (Fisher et al., 2017), we combine tonic-clonic seizures with tonic seizures to form a combined tonic (CT) seizure class. In summary, there are four seizure classes in total: CF, generalized non-specific (GN), absence (AB), and CT seizures (Table 1). In Appendix L, we show seizure classification results on the original eight seizure types in TUSZ.

**Data splits.** We randomly split the official TUSZ train set by patients into train and validation sets by 90/10 for model training and hyperparameter tuning, respectively, and we hold-out the official TUSZ test set for model evaluation (excluding five patients who exist in both the official TUSZ train and test sets). The train, validation, and test sets consist of distinct patients. See Appendix D for the number of preprocessed EEG clips and patients in each split.

Table 1: Summary of data in train and test sets of TUSZ v1.5.2. used in our study.

|  | EEG Files (% Seizure) | Patients (% Seizure) | Total Duration (% Seizure) | CF Seizures (Patients) | GN Seizures (Patients) | AB Seizures (Patients) | CT Seizures (Patients) |
|---|---|---|---|---|---|---|---|
| Train Set | 4,599 (18.9%) | 592 (34.1%) | 45,174.72 min (6.3%) | 1,868 (148) | 409 (68) | 50 (7) | 48 (11) |
| Test Set | 900 (25.6%) | 45 (77.8%) | 9,031.58 min (9.8%) | 297 (24) | 114 (11) | 49 (5) | 61 (4) |

**Baselines.** To compare our DCRNN to traditional CNNs/RNNs, we include three primary baselines: (a) Dense-CNN (Saab et al., 2020), a previous state-of-the-art CNN for seizure detection, (b) LSTM (Hochreiter & Schmidhuber, 1997), and (c) CNN-LSTM (implemented following Ahmedt-Aristizabal et al. (2020)). The baselines are trained and evaluated on the same preprocessed data. Additionally, we compare our method to the reported results of two CNNs for seizure classification that use TUSZ and test on unseen patients (Asif et al., 2020; Iesmantas & Alzbutas, 2020).

**Model training.** Training for all models was accomplished using the Adam optimizer (Kingma & Ba, 2014) in PyTorch on a single NVIDIA Titan RTX GPU. Model parameters were randomly initialized for models without self-supervised pre-training, and were initialized with the pre-trained weights of the encoder for models with self-supervised pre-training. All models were run for five runs with different random seeds. Detailed hyperparameter settings are shown in Appendix E. During training, we perform data augmentation as described in Appendix F.

## 3.2 EXPERIMENTAL RESULTS

**Graph neural network performance.** Following recent studies (Asif et al., 2020; Ahmedt-Aristizabal et al., 2020; O'Shea et al., 2020; Saab et al., 2020; Covert et al., 2019), we use AUROC and weighted F1-score as the main evaluation metrics for seizure detection and classification, respectively. Table 2 (3rd–7th rows) shows the performance of our DCRNN (without self-supervised pre-training) and the baselines. Distance graph-based DCRNN (or "Dist-DCRNN") and correlation graph-based DCRNN (or "Corr-DCRNN") without self-supervised pre-training perform on par with or better than the baselines. See Appendix G for additional evaluation scores.

Moreover, DCRNN outperforms the reported results of two existing CNNs (Asif et al., 2020; Ahmedt-Aristizabal et al., 2020) on seizure classification (Table 3). For fair comparison to Asif et al. (2020), we conduct 7-class seizure classification[2] on the same 3-fold patient-wise split.

---

[2]The 7 classes are the original TUSZ seizure types excluding myoclinic seizure.

In addition, we conduct an ablation experiment to examine the effectiveness of Fourier transform in our preprocessing step for DCRNNs. We find that frequency-domain inputs substantially outperform time-domain inputs on both seizure detection and classification (Appendix B).

Table 2: Seizure detection and seizure classification results. Mean and standard deviations are from five random runs. Best non-pretrained and pre-trained mean results are highlighted in **bold**.

| Model | Seizure Detection AUROC | | Seizure Classification Weighted F1-Score | |
|---|---|---|---|---|
| | 12-s | 60-s | 12-s | 60-s |
| Dense-CNN | $0.812 \pm 0.014$ | $0.796 \pm 0.014$ | $0.576 \pm 0.101$ | $0.626 \pm 0.073$ |
| LSTM | $0.786 \pm 0.014$ | $0.715 \pm 0.016$ | $0.652 \pm 0.019$ | $0.686 \pm 0.020$ |
| CNN-LSTM | $0.749 \pm 0.006$ | $0.682 \pm 0.003$ | $0.633 \pm 0.025$ | $0.641 \pm 0.019$ |
| Corr-DCRNN w/o Pre-training | $0.812 \pm 0.012$ | $\mathbf{0.804 \pm 0.015}$ | $\mathbf{0.710 \pm 0.023}$ | $\mathbf{0.701 \pm 0.030}$ |
| Dist-DCRNN w/o Pre-training | $\mathbf{0.824 \pm 0.020}$ | $0.793 \pm 0.022$ | $0.703 \pm 0.025$ | $0.690 \pm 0.035$ |
| Corr-DCRNN w/ Pre-training | $0.861 \pm 0.005$ | $0.850 \pm 0.014$ | $0.723 \pm 0.017$ | $\mathbf{0.749 \pm 0.017}$ |
| Dist-DCRNN w/ Pre-training | $\mathbf{0.866 \pm 0.016}$ | $\mathbf{0.875 \pm 0.016}$ | $\mathbf{0.746 \pm 0.024}$ | $\mathbf{0.749 \pm 0.028}$ |

Table 3: Comparison between DCRNNs (w/o pre-training) and existing CNNs on seizure classification. Mean and standard deviations are from five random runs. Best mean results are in **bold**.

| Model | 7-Class Classification Weighted F1-Score | CF Seizure AUROC | GN Seizure AUROC | AB Seizure AUROC | CT Seizure AUROC |
|---|---|---|---|---|---|
| Asif et al. (2020) | 0.62 | - | - | - | - |
| Iesmantas & Alzbutas (2020) | - | - | 0.78 | 0.72 | - |
| Corr-DCRNN, 12-s | $0.619 \pm 0.006$ | $0.907 \pm 0.008$ | $\mathbf{0.815 \pm 0.027}$ | $0.972 \pm 0.013$ | $0.908 \pm 0.005$ |
| Dist-DCRNN, 12-s | $0.585 \pm 0.006$ | $0.896 \pm 0.011$ | $0.814 \pm 0.027$ | $\mathbf{0.983 \pm 0.008}$ | $0.890 \pm 0.014$ |
| Corr-DCRNN, 60-s | $\mathbf{0.650 \pm 0.008}$ | $0.914 \pm 0.007$ | $0.795 \pm 0.031$ | $0.971 \pm 0.020$ | $\mathbf{0.939 \pm 0.008}$ |
| Dist-DCRNN, 60-s | $0.606 \pm 0.009$ | $\mathbf{0.920 \pm 0.004}$ | $0.811 \pm 0.032$ | $0.973 \pm 0.009$ | $0.926 \pm 0.020$ |

**Self-supervised pre-training improves graph neural network performance.** To assess the effectiveness of self-supervised pre-training in improving DCRNN's performance, we compare the results of DCRNN without and with self-supervised pre-training. As shown in Table 2 (last two rows), DCRNN with self-supervised pre-training outperforms its non-pretrained counterpart on both seizure detection and classification. Notably, Dist-DCRNN with self-supervised pre-training achieves an AUROC of 0.875 for 60-s seizure detection, matching the performance of a CNN (AUROC=0.88) that was pre-trained using supervised learning on a labeled dataset that was five times larger than TUSZ (Saab et al., 2020). Importantly, the pre-trained model weights are used for both seizure detection and classification, which indicates that our self-supervised pre-training method provides good model initialization that generalizes across tasks.

Figure 2a shows the ROC curves of median DCRNNs and baselines for seizure detection. At a low false positive rate (FPR), such as 25% FPR, pre-trained Dist-DCRNN achieves 84.3% true positive rate (TPR) and pre-trained Corr-DCRNN achieves 81.7% TPR on 12-s clips. Conversely, Dense-CNN, LSTM, and CNN-LSTM only achieve 72.8%, 67.0%, and 62.5% TPRs, respectively. Figure 2b shows the confusion matrices for Dist-DCRNNs and baselines for 12-s seizure classification. Dist-DCRNN without self-supervised pre-training achieves 93% accuracy on the rare AB seizures, providing 2 points increase over the best baseline. Furthermore, Dist-DCRNN with self-supervised pre-training achieves 74% accuracy on the rare CT seizures, providing 47 points increase in accuracy over the best baseline (Dense-CNN) and 48 points increase over non-pretrained Dist-DCRNN.

Surprisingly, despite being a majority class, many GN seizures are misclassified as CF seizures (Figure 2b). A board-certified neurologist manually analyzed the EEGs of 32 misclassified test GN seizures. We find that 27 seizures are in fact focal seizures but are mislabeled as GN seizures. In contrast, only 5 seizures are indeed generalized seizures but are misclassified by our models.

**Comparison between self-supervised pre-training and transfer learning.** To compare our self-supervised pre-training strategy to traditional transfer learning approaches, we pre-train DCRNNs for seizure detection and self-supervised prediction, respectively, on a large in-house dataset (40,316 EEGs, Table 7) and finetune the models for seizure detection and classification on TUSZ. We find that self-supervised pre-training consistently outperforms transfer learning (Appendix J).

**Comparison between self-supervised pre-training and auxiliary learning.** We also investigate whether using the self-supervised task as an auxiliary task can outperform self-supervised pre-

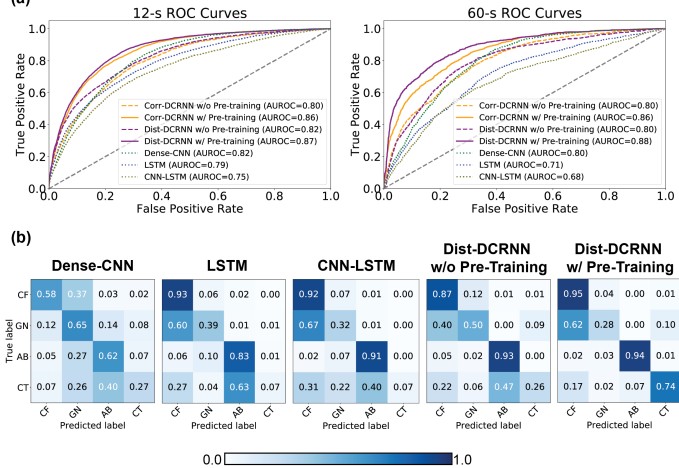

Figure 2: **(a)** ROC curves of median models for seizure detection. **(b)** Confusion matrices (averaged across five random runs) for the baselines and Dist-DCRNN without and with self-supervised pre-training for 12-s seizure classification. Each row of the confusion matrices is normalized by dividing by the number of examples in the corresponding class, such that each row sums up to one.

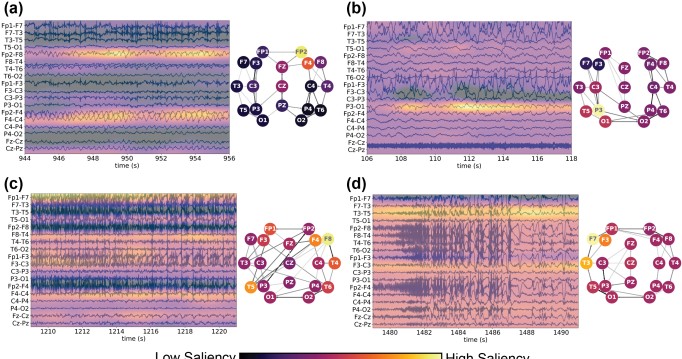

Figure 3: Occlusion maps for seizure detection obtained from Corr-DCRNN model for **(a)–(b) focal**, and **(c)–(d) generalized** seizures. In each subfigure, left panel shows the occlusion map for 12-s when a seizure occurs and right panel shows the occlusion map values averaged along time and overlaid on the corresponding correlation graph.

training. We find that auxiliary learning performs comparable to self-supervised pre-training on 12-s seizure detection, whereas self-supervised pre-training significantly outperforms auxiliary learning on 60-s seizure detection. See Appendix M and Table 10 for details.

**Improved model interpretability and ability to localize seizures.** We leverage occlusion-based techniques to localize seizures using our model predictions. Figure 3 shows example occlusion maps of correctly predicted 60-s test EEG clips from Corr-DCRNN with self-supervised pre-training. We observe that high saliency for focal seizures (Figures 3a–3b) is localized in the more abnormal EEG channels, whereas high saliency for generalized seizures (Figures 3c–3d) is more diffuse across channels. These patterns reflect the underlying brain activity of focal seizures (i.e. localized in one brain area) and generalized seizures (i.e. occur in all areas of the brain) (Fisher et al., 2017). Moreover, one can also interpret the seizure locations from the occlusion map overlaid on the graphs. In Figure 3a, Fp2, F4 and F8 are highlighted on the graph occlusion map, which correspond to the right frontal, central frontal, and anterior frontal brain regions that show the most abnormality in the EEG. In Figure 3b, while the seizure starts in P3, Fp1, F3, and F7, the graph occlusion map mainly highlights the central parietal region (P3), which is likely due to the ongoing artifact in channels Fp1-F7, Fp1-F3, and F3-C3. Figure 3c–3d show two generalized seizures that occur in all regions of the brain, and the highlighted nodes on the graphs are indeed more spread out across channels. In contrast, high saliency from Dense-CNN does not localize in any seizure regions (Appendix H).

Additionally, we leverage detailed annotations of seizure duration and location available in TUSZ dataset, and use coverage and localization scores to quantify the models' ability to accurately localize seizures. Figure 4 shows coverage and localization distributions for Dense-CNN and DCRNNs (w/o and with pre-training) on correctly predicted 60-s test EEG clips. Both Dist-DCRNN and Corr-DCRNN have many more occlusion maps with high coverage and high localization scores than Dense-CNN. This suggests that DCRNN more accurately localizes seizures than Dense-CNN, which is intuitive given that DCRNN captures the connectivity of EEG electrodes.

Clinically, localizing focal seizure onset regions is key to epilepsy surgery for patients with focal seizures. Thus, the model's ability to identify focal seizure regions as precisely as possible would be highly desirable. Notably, pre-trained Corr-DCRNN precisely localizes 25.4% focal seizures and pre-trained Dist-DCRNN precisely localizes 21.8% focal seizures (localization score > 0.8, Figure 4c). Conversely, both non-pretrained Dist-DCRNN and Corr-DCRNN precisely localize 6.3% focal seizures, and Dense-CNN only localizes 3.5% focal seizures precisely (localization score > 0.8). Figures 4e–4f show example occlusion maps of test focal seizures whose localization scores > 0.9 from pre-trained Corr-DCRNN, where high saliency overlaps well with annotated seizure regions.

Lastly, interpretability experiments on seizure classification show that high saliency EEG channels correspond to where the seizures are localized in CF seizures (Appendix I). This again suggests that our method could inform seizure locations for treatment strategy if used clinically.

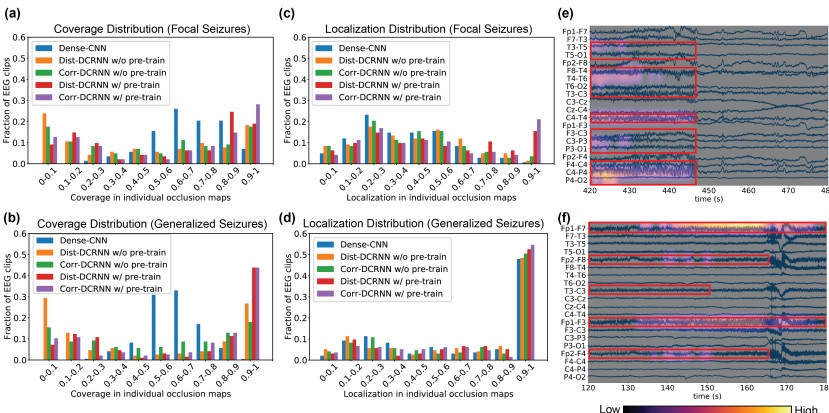

Figure 4: Distributions of **(a)–(b) coverage** and **(c)–(d) localization** scores for Dense-CNN and DCRNNs on correctly predicted 60-s EEG clips for focal and generalized seizures. **(e)–(f)** Example occlusion maps with focal seizure whose localization >0.9 from pre-trained Corr-DCRNN. Only regions with normalized occlusion value >0.5 are colored. Red boxes are annotated seizure regions.

**Comparison between graph structures.** While Dist-DCRNN and Corr-DCRNN perform comparable to each other on seizure detection and classification (Table 2), we observe that Corr-DCRNN better localizes focal seizures than Dist-DCRNN, particularly when combined with self-supervised pre-training (e.g., coverage and localization scores > 0.9 in Figure 4a, 4c). This suggests that the correlation-based graph structure could provide better interpretability and representation of focal seizure EEGs. Moreover, the correlation graph structure has two particular advantages: (a) it can be used even when the physical locations of electrodes are unknown and (b) it captures dynamic brain connectivity rather than replying purely on spatial sensor information, which is particularly desirable for automated seizure detection and classification models.

## 4 CONCLUSION

In conclusion, we present a novel method combining graph-based modeling and self-supervised pre-training for EEG-based seizure detection and classification, as well as an interpretability method to quantify model ability to localize seizures. Our method sets new state-of-the-art on both seizure detection and classification on a large public dataset, significantly improves classification of rare seizure classes, and more accurately localizes seizures. We also find that the correlation-based graph more accurately localizes focal seizures than distance-based graph. The improved ability to localize seizures and the novel graph visualizations could provide clinicians with valuable insights about localized seizure regions in real-world clinical settings. Looking to the future, because our methods are not confined to EEG alone, our study opens exciting opportunities to build graph-based representations for a wide variety of medical time series.

## ACKNOWLEDGMENTS

This work was supported by a Wu Tsai Neurosciences Institute Neuroscience Translate Grant. The authors would like to thank Bibek Paudel, Liangqiong Qu, Jean Benoit Delbrouck, and Nandita Bhaskhar for their feedback on the paper.

## ETHICS STATEMENT

The Temple University Hospital EEG Seizure Corpus used in our study is anonymized and publicly available[3] with full IRB approval (Shah et al., 2018; Obeid & Picone, 2016). No conflict of interest is reported from the authors. No harmful insights are provided by the seizure detection and classification models described in this study. Although we show that our methods could provide improved performance and clinical utility for seizure detection and classification, additional model validations are needed before they can be used in real-world clinical settings, including (a) validation on datasets from multiple institutions, (b) validation on different populations and age groups, and (c) validation on a EEG waveform-based seizure classification scheme that is approved by a consensus of board-certified neurologists.

## REPRODUCIBILITY STATEMENT

The Temple University Hospital EEG Seizure Corpus used in our study is publicly available. Detailed data preprocessing steps are provided in Appendix A. Source code is publicly available at `https://github.com/tsy935/eeg-gnn-ssl`.

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

APPENDIX

## A    DATA PREPROCESSING

Because the EEG signals are sampled at different frequencies in the Temple University EEG Seizure Corpus (TUSZ), we resample them to the same frequency of 200 Hz using the "resample" function in SciPy python package (Virtanen et al., 2020a). We perform the following preprocessing steps to obtain EEG clips in the frequency domain and their corresponding labels.

First, for seizure detection, we use both seizure and non-seizure EEGs. We obtain the EEG clips by sliding a 12-s (or 60-s) window over the EEG signals without overlaps, and ignore the last window if it is shorter than the clip length. The label for each clip is $y = 1$ if at least one seizure event occurs within this clip, otherwise $y = 0$.

Second, for seizure classification, we use only seizure EEGs. We obtain one 12-s (or 60-s) EEG clip for each seizure event starting at 2-s before the annotated seizure onset time, where a 2-s offset accounts for tolerance in the annotations. If a seizure event is shorter than 12-s (or 60-s), the EEG clip is truncated at the end of the seizure to prevent a clip from having multiple seizure types. The label for each clip is the index of the corresponding seizure class, i.e. $y \in \{0, 1, 2, 3\}$, which corresponds to combined focal (CF) seizures, generalized non-specific (GN) seizures, absence (AB) seizures, and combined tonic (CT) seizures.

Third, for self-supervised pre-training, we use the same EEG clips as seizure detection.

Fourth, for each EEG clip in each of the seizure detection/seizure classification/self-supervised pre-training tasks, we perform the following preprocessing steps to transform the signals in time domain to frequency domain: (a) slide a $t$ second window over the EEG clip without overlap, where $t$ is the time step size for networks involving recurrent layers; (b) apply fast Fourier transform (FFT) to each $t$ second window using the "fft" function in Scipy python package (Virtanen et al., 2020b), and retain the log amplitudes of the non-negative frequency components similar to prior studies (Asif et al., 2020; Ahmedt-Aristizabal et al., 2020; Covert et al., 2019); (c) z-normalize the EEG clip with respect to the mean and standard deviation of the training data. Because EEG clips for seizure classification may have variable lengths due to short seizures, we pad the clips with 0's to facilitate model training in batches. We use $t = 1$ second as a natural choice of the time step size. After preprocessing, each EEG clip can be denoted as $\boldsymbol{X} \in \mathbb{R}^{T \times N \times M}$, where $T = 12$ (or $T = 60$) represents the clip length, $N = 19$ represents the number of EEG channels/electrodes, and $M = 100$ represents the feature dimension after the aforementioned Fourier transform.

## B    EFFECTIVENESS OF FOURIER TRANSFORM

To evaluate the effectiveness of Fourier transform (Appendix A), we compare the performance of our DCRNN (without self-supervised pre-training) on inputs without and with Fourier transform on both seizure detection and seizure classification. As shown in Table 4, frequency-domain inputs (with Fourier transform) result in significantly better performance than time-domain inputs (without Fourier transform). This is likely because seizures are associated with electrical activity in certain frequency bands (Tzallas et al., 2009), and thus short-time-interval frequency-domain inputs could be more informative than time-domain inputs.

## C    DISTINGUISHABILITY OF FOCAL NON-SPECIFIC, SIMPLE PARTIAL, AND COMPLEX PARTIAL SEIZURES

Because simple partial and complex partial seizures are focal seizures characterized by the clinical behavior, consciousness during seizure (Fisher et al., 2017), they are not distinguishable from other focal seizures from EEG signals alone. In this study, we merge focal non-specific (FN), simple partial (SP), and complex partial (CP) seizures into a combined focal seizure class. To justify our decision, we perform classification of these focal seizure types using our DCRNNs and baselines. As shown in Figure 5, SP and CP seizures are largely misclassified as FN seizures, suggesting that

Table 4: Seizure detection and classification results from DCRNNs (without self-supervised pre-training) on time-domain inputs and frequency-domain inputs. Mean and standard deviations are obtained from five random runs.

| Model | Input Domain | Seizure Detection AUROC | | Seizure Classification Weighted F1-Score | |
|---|---|---|---|---|---|
| | | 12-s | 60-s | 12-s | 60-s |
| Corr-DCRNN Without Pre-Training | Time | $0.717 \pm 0.003$ | $0.704 \pm 0.023$ | $0.597 \pm 0.023$ | $0.618 \pm 0.018$ |
| | Frequency | $\mathbf{0.812 \pm 0.012}$ | $\mathbf{0.804 \pm 0.015}$ | $\mathbf{0.710 \pm 0.023}$ | $\mathbf{0.701 \pm 0.030}$ |
| Dist-DCRNN Without Pre-Training | Time | $0.733 \pm 0.012$ | $0.698 \pm 0.003$ | $0.592 \pm 0.022$ | $0.608 \pm 0.015$ |
| | Frequency | $\mathbf{0.825 \pm 0.019}$ | $\mathbf{0.793 \pm 0.022}$ | $\mathbf{0.703 \pm 0.025}$ | $\mathbf{0.690 \pm 0.035}$ |

machine learning models may not be able to distinguish these focal seizure types from EEG signals alone. This supports our decision of combining FN, SP, and CP seizures into one class.

## D  TRAIN, VALIDATION, AND TEST SETS

Table 5 shows the number of EEG clips and number of patients in our train, validation, and test sets for self-supervised pre-training, seizure detection, and seizure classification tasks. Train, validation, and test sets consist of distinct patients.

Table 5: Number of EEG clips and patients in the train, validation, and test sets in our study. Train, validation, and test sets consist of distinct patients.

| | EEG Clip Length (Secs) | Train Set | | Validation Set | | Test Set | |
|---|---|---|---|---|---|---|---|
| | | EEG Clips (% Seizure) | Patients (% Seizure) | EEG Clips (% Seizure) | Patients (% Seizure) | EEG Clips (% Seizure) | Patients (% Seizure) |
| Pre-training & Seizure Detection | 60-s | 38,613 (9.3%) | 530 (34.0%) | 5,503 (11.4%) | 61 (36.1%) | 8,848 (14.7%) | 45 (77.8%) |
| | 12-s | 196,646 (6.9%) | 531 (33.9%) | 28,057 (8.7%) | 61 (36.1%) | 44,959 (10.9%) | 45 (77.8%) |
| Seizure Classification | 60-s & 12-s | 1,925 (100.0%) | 179 (100.0%) | 450 (100.0%) | 22 (100.0%) | 521 (100.0%) | 34 (100.0%) |

## E  DETAILS OF MODEL TRAINING PROCEDURES AND HYPERPARAMETERS

We performed the following hyperparameter search on the validation set: (a) initial learning rate within range [5e-5, 1e-3]; (b) $\tau \in \{2, 3, 4\}$, the number of neighbors to keep for each node in the correlation graphs; (c) the number of Diffusion Convolutional Gated Recurrent Units (DCGRU) layers within range $\{2, 3, 4, 5\}$ and hidden units within range $\{32, 64, 128\}$; (d) the maximum diffusion step $K \in \{2, 3, 4\}$; (e) dropout probability in the last fully connected layer. We used a batch size of 40 EEG clips, the maximum possible across all models and baselines on a single Titan RTX GPU. The hyperparameters were selected based on the best performance on the validation set, and are detailed in the next sections. We used the cosine annealing learning rate scheduler (Loshchilov & Hutter, 2017) in PyTorch for all model training. We ran five runs with different random seeds for all models. In all experiments, model training was early stopped when the validation loss did not decrease for five consecutive epochs.

**Model training for seizure detection.** For seizure detection, the plentiful negative examples in the train set were undersampled such that the train set had 50% positive examples, which resulted in 27,292 training examples for 12-s clips and 7,188 training examples for 60-s clips. We used binary cross-entropy as the loss function to train the seizure detection models. The models were trained for 100 epochs with an initial learning rate of 1e-4. For the correlation graphs, the top-3 neighbors' edges were kept for each node. The maximum number of diffusion step was 2, and the dropout probability was 0 (i.e. no dropout). The model consists of two stacked DCGRU layers with 64 hidden units, resulting in 168,641 trainable parameters for the distance graph and 280,769 trainable

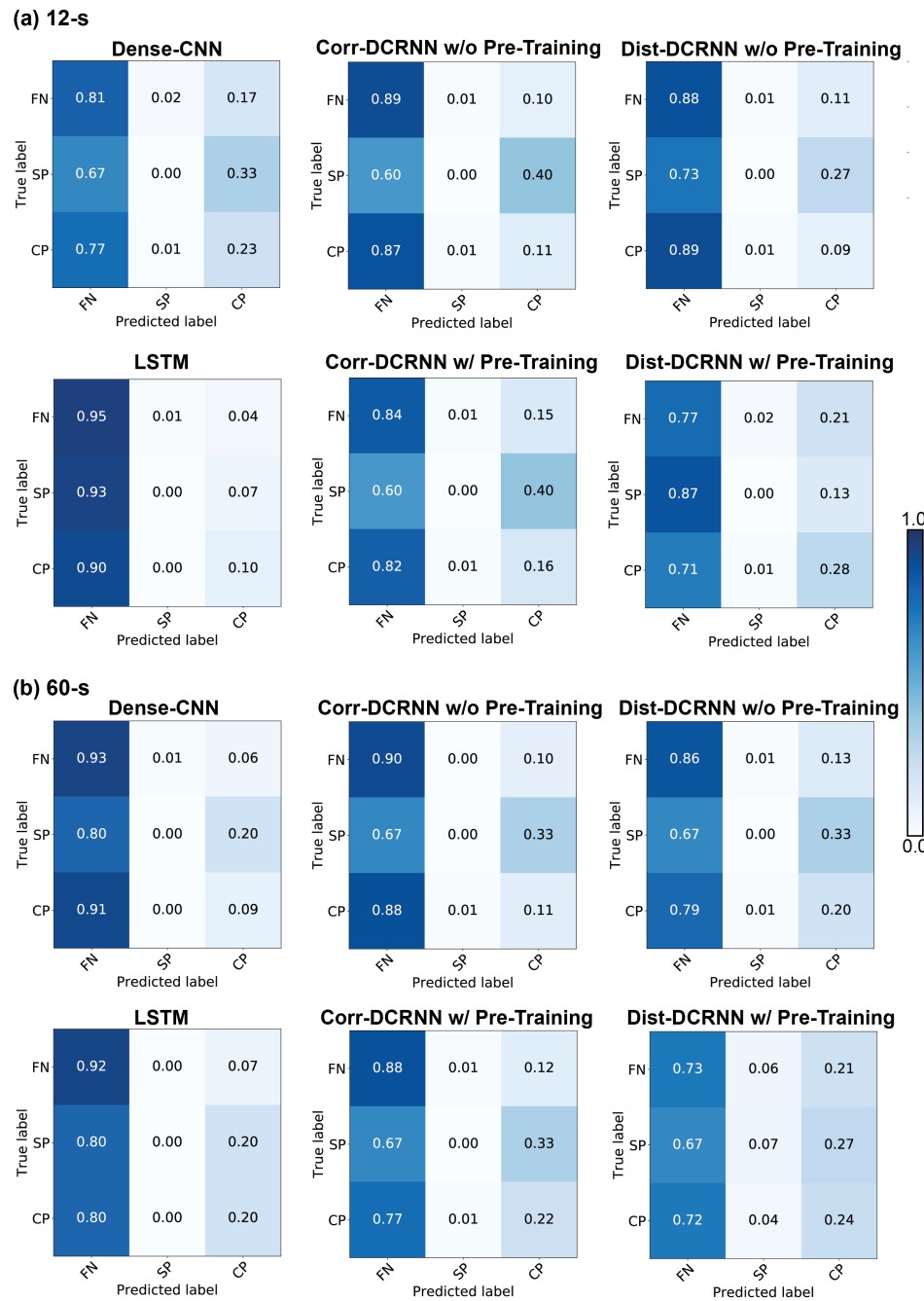

Figure 5: Confusion matrices (averaged across five random runs) of focal seizure classification on **(a) 60-s EEG clips**, and **(b) 12-s EEG clips**. All models fail to distinguish well among these focal seizure types, supporting our decision of merging these seizure types into one combined focal seizure class. Note that each row of the confusion matrices is normalized by dividing by the number of examples in the corresponding class, such that each row sums up to one.

parameters for the correlation graphs. Model training for seizure detection took about 20-min for 12-s EEG clips, and about 30-min for 60-s EEG clips.

To obtain the final seizure/non-seizure prediction, we performed decision threshold search on the validation set. More specifically, to balance between precision and recall scores, we selected the

decision threshold that results in the highest F1-score on the validation set. When evaluating the models on the test set, EEG clips with probabilities above this decision threshold are predicted as seizures, while clips with probabilities below this decision threshold are predicted as non-seizures.

**Model training for seizure classification.** For seizure classification, we used multi-class cross-entropy as the loss function during training. The models were trained for 60 epochs with an initial learning rate of 3e-4. For the correlation graphs, the top-3 neighbors' edges were kept for each node. The maximum number of diffusion step was 2, and the dropout probability was 0.5. The model consists of two stacked DCGRU layers with 64 hidden units, resulting in 168,836 trainable parameters for the distance graph and 280,964 trainable parameters for the correlation graphs. Model training for seizure classification took about 3-min for 12-s EEG clips, and about 7-min for 60-s EEG clips.

**Model training for self-supervised task.** Preliminary experiments suggested that predicting future $T' = 12$ second preprocessed EEG clips results in low regression loss on the validation set given previous 12-s (60-s) preprocessed clips, and thus we used $T' = 12$ in all self-supervised pre-training experiments. For self-supervised pre-training, we used mean absolute error (MAE) as the loss function. The models were trained for 350 epochs with an initial learning rate of 5e-4. For the correlation graphs, the top-3 neighbors' edges were kept for each node. The maximum number of diffusion step was 2. The model consists of three stacked DCGRU layers with 64 hidden units in both the encoder and decoder, resulting in 417,572 trainable parameters for the distance graph and 690,980 trainable parameters for the correlation graphs. Model training for self-supervised prediction took about 10-h for 12-s EEG clips, and about 24-h for 60-s EEG clips.

**Model training for baselines.** For baseline Dense-CNN, we employ the same model architecture as that described in Saab et al. (2020). For baseline LSTM (Hochreiter & Schmidhuber, 1997), we have the number of LSTM layers and hidden units the same as the number of DCGRU layers and hidden units in our DCRNN model. For baseline CNN-LSTM, we use the same model architecture described in Ahmedt-Aristizabal et al. (2020), i.e., two stacked convolutional layers (32 $3 \times 3$ kernels), one max-pooling layer ($2 \times 2$), one fully-connected layer (output neuron = 512), two stacked LSTM layers (hidden size = 128), and one fully connected layer.

# F    DATA AUGMENTATION

During training, we performed the following data augmentations based on EEG domain knowledge: (a) randomly scaling the amplitude of the raw EEG signals by a scale within [0.8, 1.2] and (b) randomly reflecting the signals along the scalp midline.

# G    ADDITIONAL EVALUATION RESULTS ON SEIZURE DETECTION

Table 6 shows F1-score, Area Under the Precision-Recall Curve (AUPR), sensitivity, and specificity of seizure detection models.

# H    SEIZURE DETECTION OCCLUSION MAPS FROM BASELINE DENSE-CNN

Figure 6 shows example seizure detection occlusion maps obtained from baseline Dense-CNN on correctly predicted 60-s EEG clips in the test set. Unlike our model (Figure 3), high saliency from Dense-CNN does not localize in any seizure regions.

# I    SEIZURE CLASSIFICATION OCCLUSION MAPS FROM DCRNN

Figure 7 shows example seizure classification occlusion maps from Corr-DCRNN with self-supervised pre-training on correctly predicted 60-s EEG clips in the test set. The occlusion maps are obtained by completely dropping one EEG channel at a time and calculating the relative change in the model output. For CF seizures (a–b), high saliency regions correspond to brain areas where the focal seizures are localized. For the other generalized seizure types (c–e), less salient regions correspond to less abnormal areas.

Table 6: Additional evaluation scores for seizure detection on **(a)** 12-s EEG clips and **(b)** 60-s EEG clips. Mean and standard deviations are obtained from five random runs. Best non-pretrained and pre-trained mean results are highlighted in **bold**.

(a) 12-s EEG clips

| Model | F1-Score (mean ± std) | AUPR (mean ± std) | Sensitivity (mean ± std) | Specificity (mean ± std) |
|---|---|---|---|---|
| Dense-CNN | 0.326 ± 0.019 | 0.328 ± 0.043 | 0.293 ± 0.021 | 0.938 ± 0.014 |
| LSTM | 0.376 ± 0.021 | 0.354 ± 0.023 | 0.357 ± 0.045 | 0.934 ± 0.015 |
| CNN-LSTM | 0.337 ± 0.009 | 0.309 ± 0.015 | 0.333 ± 0.028 | 0.920 ± 0.021 |
| Corr-DCRNN Without Pre-training | 0.392 ± 0.027 | 0.370 ± 0.027 | 0.373 ± 0.035 | 0.935 ± 0.012 |
| Dist-DCRNN Without Pre-training | **0.437 ± 0.029** | **0.411 ± 0.041** | **0.411 ± 0.038** | **0.943 ± 0.006** |
| Corr-DCRNN With Pre-training | 0.484 ± 0.011 | 0.454 ± 0.020 | 0.524 ± 0.012 | 0.922 ± 0.004 |
| Dist-DCRNN With Pre-training | **0.487 ± 0.042** | **0.463 ± 0.048** | **0.592 ± 0.052** | 0.897 ± 0.012 |

(b) 60-s EEG clips

| Model | F1-Score (mean ± std) | AUPR (mean ± std) | Sensitivity (mean ± std) | Specificity (mean ± std) |
|---|---|---|---|---|
| Dense-CNN | 0.404 ± 0.022 | 0.399 ± 0.017 | 0.451 ± 0.134 | 0.869 ± 0.071 |
| LSTM | 0.365 ± 0.009 | 0.287 ± 0.026 | **0.463 ± 0.060** | 0.814 ± 0.053 |
| CNN-LSTM | 0.330 ± 0.016 | 0.276 ± 0.009 | 0.363 ± 0.044 | 0.857 ± 0.023 |
| Corr-DCRNN Without Pre-training | **0.448 ± 0.029** | **0.440 ± 0.021** | 0.457 ± 0.058 | 0.900 ± 0.028 |
| Dist-DCRNN Without Pre-training | 0.341 ± 0.170 | 0.418 ± 0.046 | 0.326 ± 0.183 | **0.932 ± 0.058** |
| Corr-DCRNN With Pre-training | 0.514 ± 0.028 | 0.539 ± 0.024 | 0.502 ± 0.047 | 0.923 ± 0.008 |
| Dist-DCRNN With Pre-training | **0.571 ± 0.029** | **0.593 ± 0.031** | **0.570 ± 0.047** | **0.927 ± 0.012** |

## J  COMPARISON BETWEEN SELF-SUPERVISED PRE-TRAINING AND TRANSFER LEARNING

To compare our self-supervised pre-training strategy to traditional transfer learning, we perform transfer learning by pre-training DCRNNs for seizure detection on an in-house dataset (40,316 EEGs, Table 7) and finetuning on TUSZ data for seizure detection and classification. Due to the lack of fine-grained seizure type labels, we do not pre-train DCRNNs for seizure classification on the in-house dataset. Moreover, we pre-train DCRNNs on the in-house dataset using our self-supervised pre-training strategy and finetuned them for seizure detection and classification on TUSZ.

Table 8 shows DCRNN results with self-supervised pre-training and transfer learning from the in-house dataset. Self-supervised pre-training from the in-house dataset (3rd-4th rows) consistently outperforms trasnfer learning from the in-house dataset (5th-6th rows). We speculate that transfer learning does not perform comparably to self-supervised pre-training because it suffers from distribution shift in the data (i.e., the in-house dataset comes from a different population and uses a slightly different EEG acquisition protocol). In contrast, by learning to predict the EEGs for the next

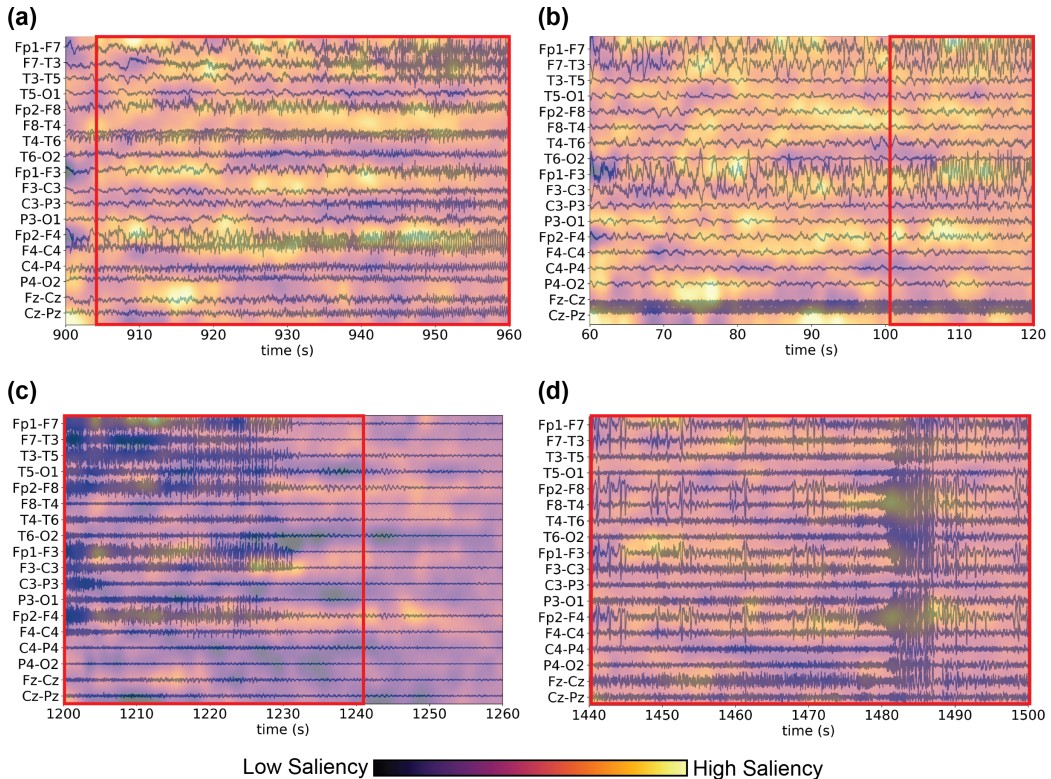

Figure 6: Example occlusion maps for seizure detection obtained from baseline Dense-CNN on correctly predicted 60-s clips for **(a)–(b) focal seizures**, and **(c)–(d) generalized seizures**. Red boxes indicate the duration of the seizures. Note that the values within an occlusion map are normalized, and thus should not be compared across different occlusion maps.

time period, self-supervised pre-training encourages the model to learn task-agnostic representations and thus mitigates the problem of distribution shift.

Table 7: Summary of in-house dataset. Only annotations for the start of seizure are available. For EEG files with seizures, EEG clips are obtained by taking 12-s (or 60-s) signals starting from the annotated seizure start time. For EEG files without seizures, EEG clips are obtained by taking 12-s (or 60-s) signals randomly from the entire signal. We apply the same data preprocessing steps described in Appendix A.

|  | EEG Files (% Seizure) | Total Duration | Patients (% Seizure) | EEG Clips (% Seizure) |
|---|---|---|---|---|
| In-House Train Set | 40,316 (24.1%) | 853,141.87 min | 5,355 (25.7%) | 46,613 (34.4%) |

## K    CHOICE OF DISTANCE-BASED GRAPH STRUCTURE

For the distance graph, we used a threholded Gaussian kernel (Shuman et al., 2013) to compute the edge weight between two electrodes (Section 2.2.1). We experimented with $\kappa$, the threshold for graph sparsity, within range [0.1, 2]. Based on preliminary experiments and EEG domain knowledge, we chose $\kappa = 0.9$ because it results in a reasonable graph (e.g. no long-range connection) that resembles the EEG montage (longitudinal bipolar and transverse bipolar) widely used clinically (Acharya et al., 2016). Figure 8 shows distance graphs resulting from different thresholds $\kappa$. In Figure 8, we can see that a smaller threshold (e.g., 0.7 or 0.8) results in missing edges between

Table 8: Comparison of results between self-supervised pre-training and transfer learning pre-trained on an in-house dataset. **3rd-4th rows**: DCRNN results with self-supervised (SS) pre-training on the in-house dataset. **5th-6th rows**: DCRNN results with transfer learning pre-trained for seizure detection on the in-house dataset. Mean and standard deviations are from five random runs. Best mean results for each column are highlighted in **bold**.

| Model | Seizure Detection AUROC | | Seizure Classification Weighted F1-Score | |
|---|---|---|---|---|
| | 12-s | 60-s | 12-s | 60-s |
| Corr-DCRNN w/ SS Pre-Training | $0.863 \pm 0.005$ | $0.856 \pm 0.013$ | $0.736 \pm 0.007$ | $\mathbf{0.723 \pm 0.010}$ |
| Dist-DCRNN w/ SS Pre-Training | $\mathbf{0.879 \pm 0.006}$ | $\mathbf{0.886 \pm 0.006}$ | $\mathbf{0.767 \pm 0.038}$ | $0.718 \pm 0.018$ |
| Corr-DCRNN w/ Transfer Learning | $0.848 \pm 0.018$ | $0.850 \pm 0.002$ | $0.733 \pm 0.027$ | $0.711 \pm 0.025$ |
| Dist-DCRNN w/ Transfer Learning | $0.866 \pm 0.001$ | $0.847 \pm 0.004$ | $0.720 \pm 0.046$ | $0.691 \pm 0.038$ |

nearby electrodes. For example, there is no edge between FP1 and FZ for threshold 0.8 and no edge between T3 and C3 for threshold 0.7. In contrast, a larger threshold (e.g., 1.0, 1.1, or 1.2) results in edges connecting electrodes that are spatially far apart. For example, there is an edge connecting C3 and FZ, as well as an edge between F7 and T5 for threshold 1.2. Using EEG domain knowledge provided by a board certified neurologist, we believe that a threshold of 0.9 results in a more reasonable distance-based EEG graph compared to other thresholding values.

In addition, we explore using a Gaussian kernel with a specified bandwidth for building the distance graph, i.e., $\boldsymbol{W}_{ij} = \frac{1}{\sqrt{2\pi h^2}} \exp\left(-\frac{\text{dist}(v_i, v_j)^2}{2h^2}\right)$, where $\boldsymbol{W}_{ij}$ is the edge weight between electrodes $v_i$ and $v_j$, $\text{dist}(v_i, v_j)$ is the Euclidean distance between $v_i$ and $v_j$, and $h$ is the Gaussian kernel bandwidth. Figure 9 shows the distance graph structures resulting from different values of bandwidth. With bandwidth = 0.06, the distance graph structure resembles the original graph structure using thresholded Gaussian kernel with a threshold of 0.9 (Figure 1b).

## L  8-CLASS SEIZURE CLASSIFICATION

We also perform seizure classification on the original eight seizure types[4] available in TUSZ (see Table 9). Note that only one patient's two myoclonic seizures are available in the official TUSZ train set and only one patient's one myoclonic seizure is available in the official TUSZ test set. Hence, patient-wise train/validation splits for myoclonic seizure is not possible, and we randomly assign one myoclonic seizure in the official TUSZ train set to our train split and the other to our validation split. As shown in Table 9, DCRNNs consistently outperform the baselines on 8-class seizure classification.

## M  USING SELF-SUPERVISED PREDICTION AS AN AUXILIARY TASK

Here, instead of pre-training DCRNNs for the self-supervised prediction task (i.e., predicting pre-processed EEG clips for the next time period), we conduct experiments using the self-supervised prediction task as an auxiliary task. Because EEG clips for the seizure classification task have variable lengths, we only perform this experiment for seizure detection to facilitate training in batches. Specifically, the loss function is $\mathcal{L} = \mathcal{L}_{\text{seizure detection}} + \lambda \mathcal{L}_{\text{SS prediction}}$ where $\lambda$ is a hyperparameter balancing the seizure detection loss and the self-supervised loss and is tuned on the validation set. For 12-s (or 60-s) seizure detection, the auxiliary task is to predict the next 6-s preprocessed EEG clips given the first 6-s (or 30-s) EEG clips.

---

[4] The eight seizure types are: focal seizure, generalized non-specific seizure, simple partial seizure, complex partial seizure, absence seizure, tonic seizure, tonic-clonic seizure, and myoclonic seizure.

Table 9: Results (weighted F1-scores) of seizure classification on original 8 seizure types in TUSZ. Mean and standard deviations are from five random runs. Best mean results for both non-pretrained and pretrained models are highlighted in **bold**.

| Model | 12-s | 60-s |
|---|---|---|
| Dense-CNN | $0.431 \pm 0.037$ | $0.427 \pm 0.047$ |
| LSTM | $0.515 \pm 0.025$ | $0.525 \pm 0.017$ |
| CNN-LSTM | $0.489 \pm 0.036$ | $0.509 \pm 0.021$ |
| Corr-DCRNN Without Pre-training | $0.553 \pm 0.025$ | $0.577 \pm 0.028$ |
| Dist-DCRNN Without Pre-training | $\mathbf{0.570 \pm 0.027}$ | $\mathbf{0.600 \pm 0.022}$ |
| Corr-DCRNN With Pre-training | $0.582 \pm 0.014$ | $0.591 \pm 0.008$ |
| Dist-DCRNN With Pre-training | $\mathbf{0.583 \pm 0.009}$ | $\mathbf{0.607 \pm 0.017}$ |

Table 10 shows the seizure detection results for DCRNNs when the self-supervised (SS) prediction task is used as an auxiliary task during training. Compared to self-supervised pre-training (Table 2 last two rows), auxiliary learning only marginally improves Dist-DCRNN performance on 12-s seizure detection (no statistical significance), whereas self-supervised pre-training significantly outperforms auxiliary learning for 60-s seizure detection.

Table 10: Seizure detection results for DCRNNs with self-supervised pre-training (same as Table 2) and auxiliary learning. Mean and standard deviations are from five random runs. Best mean results for Dist-DCRNN/Corr-DCRNN are in **bold**.

| Model | Seizure Detection AUROC | |
|---|---|---|
| | 12-s | 60-s |
| Corr-DCRNN w/ SS Pre-Training (Table 2) | $\mathbf{0.861 \pm 0.005}$ | $\mathbf{0.850 \pm 0.013}$ |
| Corr-DCRNN w/ SS Auxiliary Task | $0.851 \pm 0.008$ | $0.811 \pm 0.005$ |
| Dist-DCRNN w/ SS Pre-Training (Table 2) | $0.866 \pm 0.016$ | $\mathbf{0.875 \pm 0.016}$ |
| Dist-DCRNN w/ SS Auxiliary Task | $\mathbf{0.875 \pm 0.005}$ | $0.840 \pm 0.013$ |

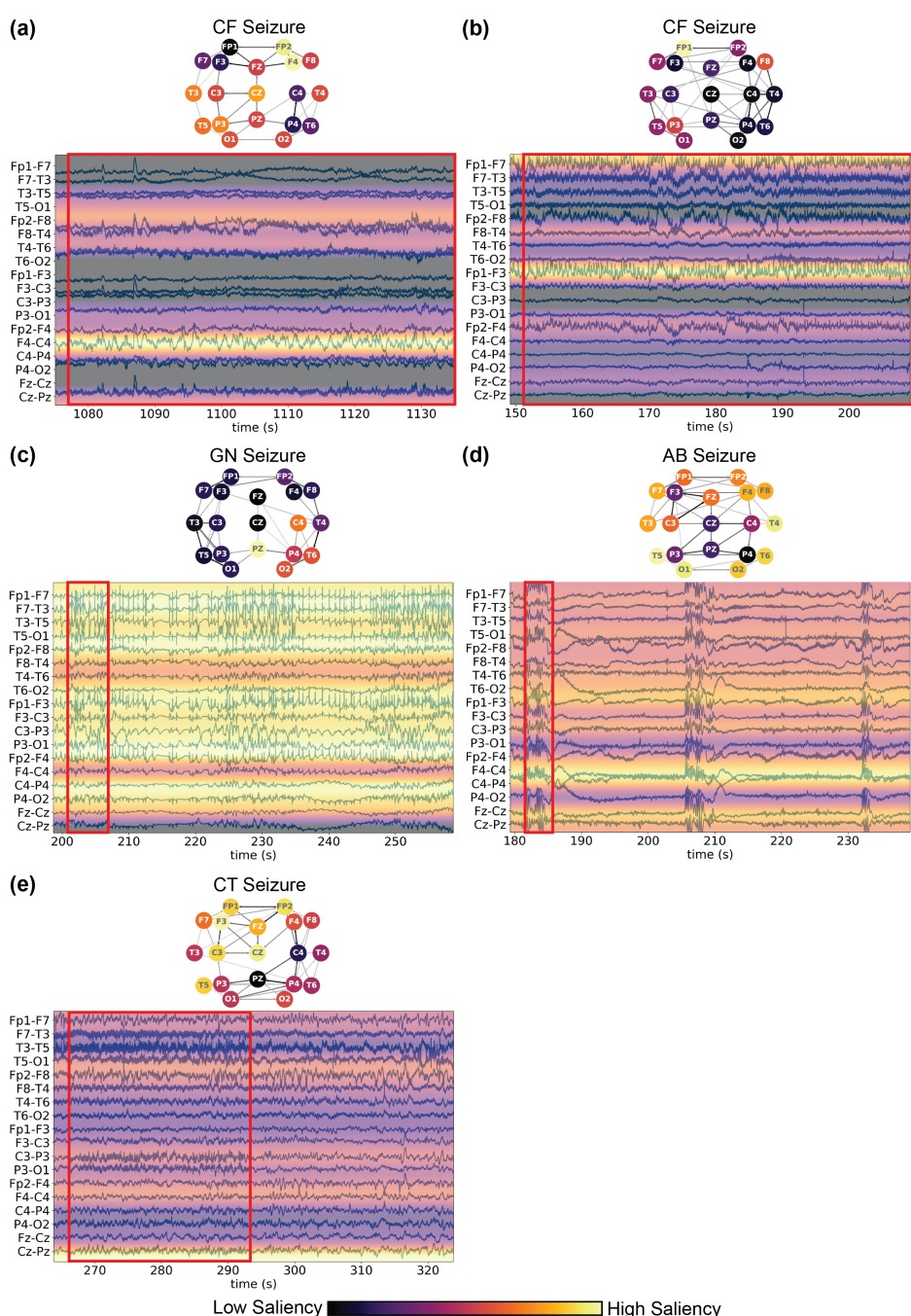

Figure 7: Example occlusion maps for seizure classification obtained from pre-trained Corr-DCRNN model for **(a)–(b)** combined focal (CF) seizures, **(c)** generalized non-specific (GN) seizure, **(d)** absence (AB) seizure, and **(e)** combined tonic (CT) seizure. In each subfigure, the bottom panel shows the occlusion values for each channel replicated along the time dimension and overlaid on 60-s EEG signals, the red boxes indicate the duration of seizures in the EEG clips, and the top panel shows the occlusion values overlaid on the correlation graph structure. To visualize occlusion map values in this "double banana" montage, we subtract the occlusion values between the corresponding channels in the montage, which results in different values between occlusion maps shown on the EEG signals (bottom panel in each subfigure) and that shown on the graph structures (top panel in each subfigure). Note that the values within an occlusion map are normalized, and thus should not be compared across different occlusion maps.

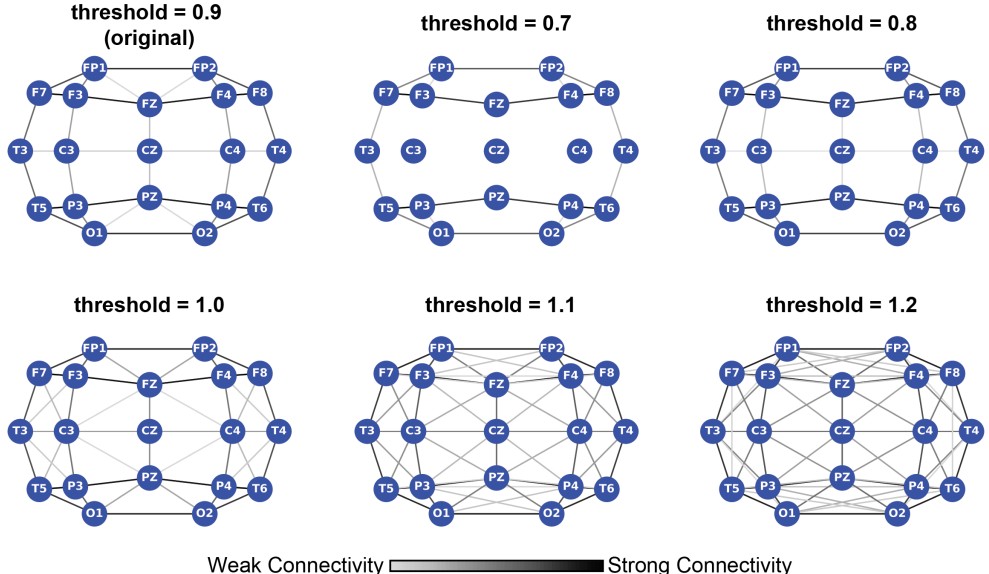

Figure 8: Distance graphs with different threshold $\kappa$ in the thresholded Gaussian kernel. Small thresholds (e.g., 0.7 and 0.8) result in missing edges between nearby electrodes, whereas large thresholds (e.g., 1.0, 1.1, 1.2) result in edges connecting electrodes that are spatially far away.

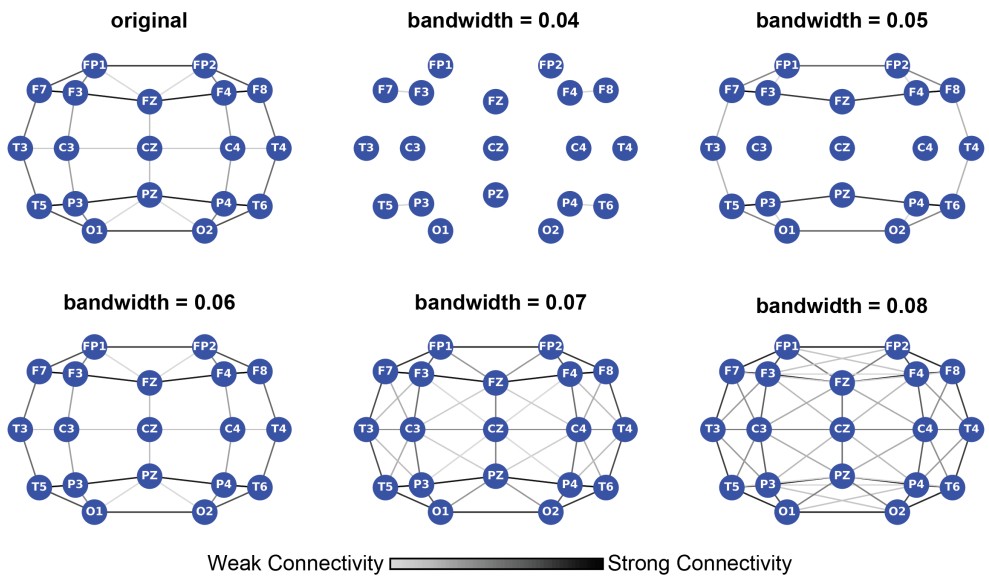

Figure 9: Distance-based graphs constructed using Gaussian kernels with different bandwidths. When bandwidth = 0.06, the distance graph structure resembles the original graph structure using thresholded Gaussian kernel with a threshold of 0.9 (Figure 1b).

