# OpenReview forum: "Self-Supervised Graph Neural Networks for Improved Electroencephalographic Seizure Analysis"
_ICLR.cc/2022/Conference — ICLR 2022 Poster_

### Official Review · Reviewer_6FoL · 2021-10-31

**Correctness:** 3
**Technical Novelty And Significance:** 3
**Empirical Novelty And Significance:** 3
**Recommendation:** 8
**Confidence:** 4

**Main Review:**

Pros.
-The paper combines well-founded strategies for representing EEG data on non-euclidean spaces preserving spatial and temporal patterns.
-The algorithm can predict and localize seizures.
-The mathematical background is limited but enough to clarify the main paper insights.
-The experiments are explained in detail and include ablation procedures.
-Code is available.

Cons. and comments
-Though the autoencoder-based strategy seems helpful, It could be used as a regularization strategy within an end-to-end training. However, using the autoencoder just for initializing the network weights sounds limited; maybe other initialization and transfer learning strategies could obtain better results.
-It is not clear to me how a simple correlation can enhance a nonlinear Gaussian kernel. I think the authors need to discuss better how they fix the kernel hyperparameters. For example, you are working on a log-spectral space, and you set the Gaussian bandwidth as the standard deviation of the Euclidean distances. Still, you then impose a manifold-based affinity by thresholding.
-Authors claim that the non-Euclidean representation (affinity matrix thresholding) favors the EEG analysis. Nonetheless, if you use a Gaussian kernel with a proper bandwidth, you can obtain similar results compared to the thresholding approach without the need to fix such a hyperparameter. So then, is the "non-Euclidean" representation needed? Maybe, a good bandwidth with a good network will be enough.
-Notations in section 2.2.2, Graph neural network, use the EEG trial X as input; however, according to fig 1, the DCRNNs are applied to the affinity matrix. Can you clarify, please?

**Summary Of The Paper:**

The authors propose a graph-based representation from thresholded  Gaussian and linear (correlation) kernels (undirected connectivity) coupled with a diffusion convolutional recurrent network. Besides, a Fourier-based preprocessing is carried out with self-supervised (autoencoders) to initialize the network weights.  Experiments are performed on an EEG-based seizure detection and localization task. The seizure localization is conducted using occlusion and dropping approaches on EEG channels. Obtained results elucidate an interesting strategy for seizure analysis on a well-known public database.

**Summary Of The Review:**

Overall, the paper covers a research field of interest from theoretical (Spatio-temporal coding of time series using graph and DCRNN) and practical (EEG-based seizure detection and localization) perspectives. The experiments are exhaustive and convincing.

---

> ### Author Response · Authors · 2021-11-19
> **Response to Reviewer 6FoL (part 1)**
>
> We thank the reviewer for the positive review and insightful questions. Our responses to specific questions are listed below.
>
> 1) _Though the autoencoder-based strategy seems helpful, It could be used as a regularization strategy within an end-to-end training._
>
> Thank you for the suggestion. We agree with the reviewer that the autoencoder-based strategy could be used as a regularization within an end-to-end training. We have performed the suggested experiment using the self-supervised prediction task as an auxiliary task. Because EEG clips for the seizure classification task have variable lengths, we only perform this experiment for seizure detection to facilitate training in batches. For 12-s (or 60-s) seizure detection, the auxiliary task is to predict the next 6-s preprocessed EEG clips given the first 6-s (or 30-s) EEG clips. We have added the results of auxiliary learning in Table 10. Compared to self-supervised pre-training (Table 2 last two rows), auxiliary learning only marginally improves Dist-DCRNN performance for 12-s seizure detection (no statistical significance), whereas self-supervised pre-training significantly outperforms auxiliary learning for 60-s seizure detection. We have added details about this experiment in Section 3.2 “Comparison between self-supervised pre-training and auxiliary learning” and Appendix L.
>
> 2) _However, using the autoencoder just for initializing the network weights sounds limited; maybe other initialization and transfer learning strategies could obtain better results._
>
> Thank you for the suggestions. We agree with the reviewer that transfer learning could be an alternative to self-supervised pre-training to improve the model performance. We have added additional experiments comparing self-supervised pre-training to transfer learning in Section 3.2 and Appendix I. For transfer learning, we pre-trained our models for seizure detection on a large in-house dataset (40,316 EEGs; Table 7 in Appendix) and finetuned the models for seizure detection and classification on TUSZ dataset. As a comparison, we also pre-trained our models using our self-supervised pre-training strategy on the in-house dataset and finetuned for seizure detection and classification on TUSZ. As shown in Table 8, DCRNNs with self-supervised pre-training consistently outperforms DCRNNs with transfer learning. We speculate that transfer learning does not perform comparably to self-supervised pre-training because transfer learning suffers from distribution shift in the data (i.e., the in-house dataset comes from a different population and uses a slightly different EEG acquisition protocol). In contrast, by learning to predict the EEGs for the next time period, self-supervised pre-training encourages the model to learn task-agnostic representations and thus mitigates the problem of distribution shift.
>
> A common model initialization strategy in computer vision is to initialize models from ImageNet pre-trained weights. However, studies have shown that using ImageNet pre-trained weights may not outperform random initialization in both natural imaging (He et al., 2019) and medical imaging (Raghu et al., 2019). Moreover, to our knowledge, there is no ImageNet equivalent dataset in the EEG domain, which limits us to explore this initialization strategy.
>
> References:
> * He, K., Girshick, R., & Dollár, P. (2019). Rethinking imagenet pre-training. Proceedings of the IEEE/CVF International Conference on Computer Vision, 4918–4927.
> * Raghu, M., Zhang, C., Kleinberg, J., & Bengio, S. (2019). Transfusion: Understanding Transfer Learning for Medical Imaging. In arXiv [cs.CV]. arXiv. http://arxiv.org/abs/1902.07208

---

> ### Author Response · Authors · 2021-11-19
> **Response to Reviewer 6FoL (part 2)**
>
> 3) _It is not clear to me how a simple correlation can enhance a nonlinear Gaussian kernel. I think the authors need to discuss better how they fix the kernel hyperparameters._
>
> Thank you for the comment. For the distance-based EEG graph (Figure 1b), we used a thresholded Gaussian kernel to compute the edge weight based on the Euclidean distance between two electrodes in the standard 10-20 system (Section 2.2.1). We chose a threshold of 0.9 because preliminary experiments suggested that this threshold results in a reasonable graph based on our EEG domain knowledge. It is also important to note that our distance graph structure (Figure 1b) resembles the EEG montage (longitudinal bipolar and transverse bipolar) widely used clinically (Acharya et al., 2016).
>
> We have updated Section 2.2.1 and added Appendix J to describe how we chose the threshold hyperparameter (previously was in Appendix E). We have also added Figure 8 in the Appendix showing distance graphs resulting from different thresholds. In Figure 8, we can see that a smaller threshold (e.g., 0.7 or 0.8) results in missing edges between nearby electrodes. For example, there is no edge between FP1 and FZ for threshold 0.8 and no edge between T3 and C3 for threshold 0.7. In contrast, a larger threshold (e.g., 1.0, 1.1, or 1.2) results in edges connecting electrodes that are spatially far apart. For example, there is an edge connecting C3 and FZ, as well as an edge between F7 and T5 for threshold 1.2. Using EEG domain knowledge provided by a board certified neurologist, we believe that a threshold of 0.9 results in a more reasonable distance-based EEG graph compared to other thresholding values.
>
> 4) _For example, you are working on a log-spectral space, and you set the Gaussian bandwidth as the standard deviation of the Euclidean distances. Still, you then impose a manifold-based affinity by thresholding. -Authors claim that the non-Euclidean representation (affinity matrix thresholding) favors the EEG analysis. Nonetheless, if you use a Gaussian kernel with a proper bandwidth, you can obtain similar results compared to the thresholding approach without the need to fix such a hyperparameter. So then, is the "non-Euclidean" representation needed? Maybe, a good bandwidth with a good network will be enough._
>
> Thank you for the comment. First, we would like to clarify that the Gaussian kernel is only used to form the distance-based graph structure (i.e., the adjacency or affinity matrix), rather than applied to the preprocessed EEG signals (log-spectral space). For applications where edge weights in the graph are not naturally defined, such as EEGs, the Gaussian kernel is a common way to define the edge weights (Kriege et al., 2020; Shuman et al., 2013).
>
> Second, we would also like to clarify that by “non-Euclidean representation”, we refer to representing EEGs as graphs and using GNNs to model the complex relationship in graph-structured EEG data. CNN-based modeling represents EEGs (or their spectrograms) as “images”, which implies incorrect topology on EEGs. For example, using the double banana EEG montage (Figure 3 in our manuscript), while frontal and occipital regions are brain areas that are far apart, the frontal electrodes (Fp1 and Fp2) and occipital electrodes (O1 and O2) are placed adjacent in the EEG “image”, which could lead to suboptimal representation of EEGs.
>
> Third, we agree that a Gaussian kernel with a proper bandwidth would result in a similar distance graph structure as that used in our study (Figure 1b). We have added Appendix I where we explore Gaussian kernels with different values of bandwidths and visualize the corresponding distance-based graphs in Figure 9. As shown in Figure 9, a bandwidth of 0.06 results in a graph structure that resembles the original distance graph structure in our study. We would also like to note that although this method eliminates the need of setting a threshold, it introduces bandwidth as an additional hyperparameter. Therefore, we believe that both thresholded Gaussian kernel (in our study) and a Gaussian kernel with a proper bandwidth are equivalent and can be used to construct the distance-based EEG graph.
>
> References:
> * Acharya, J. N., Hani, A. J., Thirumala, P. D., & Tsuchida, T. N. (2016). American Clinical Neurophysiology Society Guideline 3: A Proposal for Standard Montages to Be Used in Clinical EEG. Journal of Clinical Neurophysiology: Official Publication of the American Electroencephalographic Society, 33(4), 312–316.
> * Kriege, N. M., Johansson, F. D., & Morris, C. (2020). A survey on graph kernels. Applied Network Science, 5(1), 1–42.
> * Shuman, D. I., Narang, S. K., Frossard, P., Ortega, A., & Vandergheynst, P. (2013). The emerging field of signal processing on graphs: Extending high-dimensional data analysis to networks and other irregular domains. IEEE Signal Processing Magazine, 30(3), 83–98.

---

> ### Author Response · Authors · 2021-11-19
> **Response to Reviewer 6FoL (part 3)**
>
> 5) _Notations in section 2.2.2, Graph neural network, use the EEG trial X as input; however, according to fig 1, the DCRNNs are applied to the affinity matrix. Can you clarify, please?_
>
> We apologize for the confusion. The DCRNN is applied to the preprocessed EEG clips X. In Figure 1d, each node (i.e., EEG electrode/channel) has a feature vector corresponding to the preprocessed EEG signals in that EEG channel.
>
> We have added clarifications in the caption of Figure 1.

---

### Official Review · Reviewer_a7bX · 2021-11-02

**Correctness:** 3
**Technical Novelty And Significance:** 3
**Empirical Novelty And Significance:** 3
**Recommendation:** 6
**Confidence:** 4

**Main Review:**

The details of the self supervised training must be added in more detail for clarity and reproducibility.

Self supervised methods are mostly suited for applications where labels are scarce, the authors should add more justification as to why they have targeted such a method when sufficient labels are available

An occlusion based method is used for analysis of the interpretability, the authors should comment if the graph structure can be used to interpret the significance of various brain regions (coming from the electrodes) in terms of the seizure.

Although the some classes are combined, it would be interesting to see the performance when all eight class labels present in the data are used for classification.

**Summary Of The Paper:**

The paper presents a method for seizure detection and classification. In particular, the method is self supervised, based on graph neural network and use EEG signals. The authors report significant performance in detection and classification, as well as provide methods for qualitative evaluation of model interpretability.

**Summary Of The Review:**

The authors present a graph neural network method for seizure detection and classification. The results are shown to improve current state-of-the-art. However some of the design choices and interpretability analysis needs more background and motivation. In particular, the details of the self supervised implementation needs more elaboration.

---

> ### Author Response · Authors · 2021-11-19
> **Response to Reviewer a7bX (part 1)**
>
> Thank you for the positive review and detailed comments. Our responses to specific comments are listed below.
>
> 1) _The details of the self supervised training must be added in more detail for clarity and reproducibility._
>
> Thank you for the comment. For self-supervised pre-training, the model is pre-trained for predicting the next T’ second preprocessed EEG clips given a previous preprocessed 12-s (60-s) EEG clip. We hypothesize that by learning to predict the EEG signals for the next time period, the model would learn task-agnostic representations and improve downstream seizure detection and classification tasks. The model has a sequence-to-sequence architecture with an encoder and a decoder, each of which has several stacked DCGRUs (Figure 1d). Mean absolute error between the true preprocessed EEG clips and the predicted clips is used as the loss function. Note that we used T’=12 second in all self-supervised pre-training experiments because preliminary experiments suggest that predicting future T'=12 second preprocessed EEG clips generally results in low regression loss on the validation set.
>
> We have added these additional details in Section 2.3. Model hyperparameters can be found in Appendix E. Code implementation can be found in Supplementary Materials.
>
> 2) _Self supervised methods are mostly suited for applications where labels are scarce, the authors should add more justification as to why they have targeted such a method when sufficient labels are available._
>
> Thank you for the comment. We originally explained the motivation behind self-supervised pre-training in the 4th paragraph of Introduction (page 2). Following the reviewer’s comment, we have added more justification in the paragraph. The motivation is mainly driven by the rarity of some seizure types. Although there are 4,599 EEG signals in the official TUSZ training set, the seizure classes are highly imbalanced, and there are only 50 absence (AB) seizures and 48 combined tonic (CT) seizures (see Table 1). Training machine learning models that can perform well on these two rare seizure types (AB and CT seizures) using fully supervised learning approaches would be challenging. Moreover, recent studies (Liu et al., 2021; Yang & Xu, 2020) have shown that self-supervised pre-training significantly improves model performance on data with imbalanced labels in the field of computer vision. Therefore, we hypothesize and empirically show that self-supervised pre-training can improve the model performance, particularly for classification of rare seizure types (Figure 2b).
>
> Moreover, a major portion of EEG signals do not have seizures. For example, while the EEG signals in the official TUSZ train set have a total duration of 45,174.72 min (Table 1), only 6.3% of the entire duration has seizures (and seizure type labels). Our self-supervised pre-training strategy allows the model to leverage the abundant non-seizure EEGs that are readily available in the dataset.
>
> We have updated the 4th paragraph of Introduction with more motivations for self-supervised pre-training.
>
> 3) _An occlusion based method is used for analysis of the interpretability, the authors should comment if the graph structure can be used to interpret the significance of various brain regions (coming from the electrodes) in terms of the seizure._
>
> Thank you for the comment. In Figure 3, we have shown the occlusion maps overlaid on correlation-based graphs. Indeed, we can interpret the significance of brain regions based on these visualizations. A board certified neurologist manually analyzed these raw EEG signals. In Figure 3a, electrodes Fp2, F4 and F8 are highlighted on the graph occlusion map, which correspond to the right frontal, central frontal, and anterior frontal brain regions that show the most abnormality in the EEG. In Figure 3b, while our manual analysis suggests that the seizure starts in P3, Fp1, F3, and F7, the graph occlusion map mainly highlights the central parietal brain region (P3), which is likely due to the ongoing artifact (spikes) in channels Fp1-F7, Fp1-F3, and F3-C3. Figure 3c and Figure 3d show two generalized seizures that occur in all regions of the brain, and the highlighted nodes on the graphs are indeed more spread out across channels.
>
> We have added these comments in Section 3.2 “Improved model interpretability and ability to localize seizures”.
>
> References:
> * Liu, H., HaoChen, J. Z., Gaidon, A., & Ma, T. (2021). Self-supervised Learning is More Robust to Dataset Imbalance. In arXiv [cs.LG]. arXiv. http://arxiv.org/abs/2110.05025
> * Yang, Y., & Xu, Z. (2020). Rethinking the Value of Labels for Improving Class-Imbalanced Learning. In arXiv [cs.LG]. arXiv. http://arxiv.org/abs/2006.07529

---

> ### Author Response · Authors · 2021-11-19
> **Response to Reviewer a7bX (part 2)**
>
> 4) _Although some classes are combined, it would be interesting to see the performance when all eight class labels present in the data are used for classification._
>
> Thank you for the question. We have added Table 9 in Appendix K showing the results of seizure classification on the original 8 seizure types available in TUSZ. Note that only one patient's two myoclonic seizures are available in the official TUSZ train set and only one patient's one myoclonic seizure is available in the official TUSZ test set. Hence, patient-wise train/validation splits for myoclonic seizure are not possible, and we randomly assign one myoclonic seizure in the official TUSZ train set to our train split and the other to our validation split. Similar to 4-class seizure classification in our main experiments, we find that DCRNNs consistently outperform the baselines on 8-class seizure classification.

---

### Official Review · Reviewer_axL1 · 2021-11-08

**Correctness:** 3
**Technical Novelty And Significance:** 3
**Empirical Novelty And Significance:** 3
**Recommendation:** 8
**Confidence:** 4

**Main Review:**

Strengths:
- The paper is very well written with the 3 goals clearly stated and how they were addressed.
- The paper proposes two graph structures to model natural geometry of EEG sensors and dynamic brain connectivity. It compares the two results and shows that correlation based structure (dynamic brain connectivity) performed better than the one based on distance (natural geometry) for a rare seizure type
- The value of pre-training was shown is very clear from experimental results.
- The accuracy of occlusion based seizure localization was quantified and visualized in a very comprehensible way. It shows the value of pre-training. It was great to see two goals of the paper (interpretability and effect of pre-training) demonstrated in one set of results.

Weaknesses:
- The paper does not include a lot of comparison with other methods but since it builds on multiple approaches (pre-training and two graph structures) there are 4 models resulting from these.
- The individual methods are built upon exising work

**Summary Of The Paper:**

The paper proposes a method to combine modeling based on graphs mapping electrode geometry and self-supervised pre-training for EEG-based seizure detection and classification. It also proposes an interpretation method using occlusion maps to the demonstrate the model's ability to localize seizure.

**Summary Of The Review:**

The paper proposes and  two graph structures to model EEG sensors and brain connectivity and a pre-training method to improve model performance. Both of these are well demonstrated and the results demonstrate how one structure is superior to another for a a particular type of rare seizure. It also includes an occlusion map based seizure localization method. Overall, the paper clearly states its goal and does a good job of reaching those goals using well demonstrated experimental results.

---

> ### Author Response · Authors · 2021-11-19
> **Response to Reviewer axL1**
>
> We thank the reviewer for the positive review and detailed comments. Responses to specific comments are listed below.
>
> 1) _Weaknesses: The paper does not include a lot of comparison with other methods but since it builds on multiple approaches (pre-training and two graph structures) there are 4 models resulting from these._
>
> Thank you for the comment. We have implemented another baseline, CNN-LSTM (2 convolutional layers → 1 max-pooling layer → 1 fully connected layer → 2 LSTM layers → 1 fully connected layer), based on the descriptions in (Ahmedt-Aristizabal et al. 2020). We observe that CNN-LSTM underperforms DCRNNs without self-supervised pre-training on all of the seizure detection and classification tasks, which is expected given that CNN-LSTM also assumes Euclidean structure in EEGs. We have updated Table 2 and Figure 2 in the manuscript with the results of CNN-LSTM.
>
> We would like to point out that most of the previous automated seizure detection or classification studies did not open source their code, and this has limited us to compare our methods to a lot of prior methods. In contrast, we have released our source code.
>
> 2) _The individual methods are built upon existing work_
>
> Thank you for the comment. While our study is built upon previous work that we have cited in detail, there are clear novelties in our work:
> * We propose and study in detail two different EEG graph structures. The distance-based graph structure represents the natural geometry of EEG electrodes and is universal across all the EEG clips. Whereas the correlation-based graph structure captures the dynamic connectivity in the brain and is unique for each EEG clip. Moreover, we show that the correlation-based graph structure is better at localizing focal seizures than the distance-based graph structure (last paragraph in Section 3.2).
> * We propose a self-supervised pre-training strategy for EEGs that greatly improve our model performance, particularly on rare seizure types. Our self-supervised pre-training method—predicting EEGs for next time period using a sequence-to-sequence architecture—is new in the EEG domain.
> * We propose a quantitative method to perform and evaluate model ability to localize seizures. In prior seizure detection/classification studies, only qualitative visualization has been shown for model interpretability (Covert et al., 2019; Saab et al., 2020). In contrast, by combining occlusion and quantitative metrics (coverage and localization), our proposed quantitative interpretability method allows us to measure how well a model localizes seizures in a principled manner.
>
> References:
> * Ahmedt-Aristizabal, D., Fernando, T., Denman, S., Petersson, L., Aburn, M. J., & Fookes, C. (2020). Neural Memory Networks for Seizure Type Classification. Conference Proceedings: ... Annual International Conference of the IEEE Engineering in Medicine and Biology Society. IEEE Engineering in Medicine and Biology Society. Conference, 2020, 569–575.
> * Covert, I. C., Krishnan, B., Najm, I., Zhan, J., Shore, M., Hixson, J., & Po, M. J. (2019). Temporal Graph Convolutional Networks for Automatic Seizure Detection. In F. Doshi-Velez, J. Fackler, K. Jung, D. Kale, R. Ranganath, B. Wallace, & J. Wiens (Eds.), Proceedings of the 4th Machine Learning for Healthcare Conference (Vol. 106, pp. 160–180). PMLR.
> * Saab, K., Dunnmon, J., Ré, C., Rubin, D., & Lee-Messer, C. (2020). Weak supervision as an efficient approach for automated seizure detection in electroencephalography. NPJ Digital Medicine, 3, 59.

---

> > ### Comment · Reviewer_axL1 · 2021-11-22
> > **Thanks for the revision**
> >
> > Thanks for the revision and diligently addressing my comments including adding the baseline. I have increased my score after reviewing the changes and going through the explanations.

---

> > > ### Author Response · Authors · 2021-11-22
> > > **Thank you**
> > >
> > > Thank you for the review!

---

### Author Response · Authors · 2021-11-19
**Response to AC and all reviewers**

Dear AC and Reviewers,

We are glad to see a high level of enthusiasm for our work. We thank the reviewers for their close read of this manuscript and their insightful comments. Several important suggestions were made and we have considered each carefully and revised accordingly. Please find our replies to the reviewers below for detailed responses to the reviewers’ comments. We have also updated our manuscript with the following changes that we believe have improved the paper:
* Implemented another baseline model (CNN-LSTM) and added its results for seizure detection and classification in Table 2, Figure 2, and Table 6 (Reviewer axL1).
* Added more justification and details about self-supervised pre-training in Introduction and Section 2.3 (Reviewer a7bX).
* Commented on interpretation of seizure brain regions from occlusion maps on graphs in Section 3.2 “Improved model interpretability and ability to localize seizures” (Reviewer a7bX).
* Added results of 8-class seizure classification (original TUSZ seizure types) in Appendix K (Reviewer a7bX).
* Conducted new experiments comparing self-supervised pre-training to transfer learning by pre-training our models on a large in-house dataset and finetuning on TUSZ dataset for seizure detection and classification. Results are detailed in Section 3.2 “Comparison between self-supervised pre-training and transfer learning” and Appendix I (Reviewer 6FoL).
* Conducted new experiments comparing self-supervised pre-training to end-to-end training using the self-supervised prediction task as an auxiliary task in Section 3.2 “Comparison between self-supervised pre-training and auxiliary learning” and Appendix L (Reviewer 6FoL).
* Explained in more details how we choose the hyperparameter when constructing the distance-based graph and showed that a Gaussian kernel with a proper bandwidth results in the same distance-based graph in Section 2.2.1 and Appendix J (Reviewer 6FoL).

---

### Decision · Program_Chairs · 2022-01-20

**Decision:**

Accept (Poster)

**Comment:**

This work tackles an important clinical application. It is experimentally solid and investigates
novel deep learning methodologies in a convincing way.

For these reasons, this work is endorsed for publication at ICLR 2022.